# Sanctuary for vulnerable Arctic species at the Borealis Mud Volcano

Giuliana Panieri [1] ✉, Claudio Argentino [1], Alessandra Savini [2],
Bénédicte Ferré [1], Fereshteh Hemmateenejad [2], Mari H. Eilertsen [3],
Rune Mattingsdal[4], Sofia P. Ramalho [5], Tor Eidvin[6], Sarah Youngs [7],
Beckett Casper Colson[7], Anna Pauline Miranda Michel [7],
Jason Alexander Kapit[7], Denise Swanborn [8], Alex D. Rogers[9,10],
Ines Barrenechea Angeles[1], Stéphane Polteau[11], Dimitri Kalenitchenko[1,12],
Stefan Buenz[1] & Adriano Mazzini [11,13]

Borealis is a recently discovered submerged mud volcano in the Polar North Atlantic, differing from the numerous methane seepages previously identified in the region. Here we show in situ observations from a remotely operated vehicle (ROV), capturing the release of warm (11.5 °C) Neogene sediments and methane-rich fluids from a gryphon at Borealis. The surrounding seafloor within the mud volcano features extensive carbonate deposits, indicating prolonged diffuse methane migration. Sampling and imagery reveal that Borealis supports unique habitats adapted to low-oxygen conditions near methane seeps. Additionally, the irregularly shaped carbonate structures serve as a natural shelter from bottom trawling and a substratum for sessile fauna and may function as nursery grounds for threatened fish species. This discovery underscores the ecological significance of cold seep ecosystems in the Polar North Atlantic, highlighting their role in biodiversity by serving as refuges for marine species and emphasizing the need for their conservation.

Over the past ten years, marine surveys in the Polar North Atlantic continental shelf and slope have consistently identified new methane seeps on the seafloor[1] (Fig. 1). These sites are of great interest because of their potential impact on the marine ecosystem and global climate. Methane, a potent greenhouse gas, has increased atmospheric concentrations since the start of the Industrial Revolution, accelerating climate change[2]. Estimates suggest that between 218 and 371 teragrams of methane per year (Tg $CH_4$ $yr^{-1}$) are emitted from natural sources within terrestrial and aquatic settings, as determined by top-down and bottom-up approaches, respectively[3]. Methane's influence on the environment extends beyond its well-known role as a greenhouse gas; it is also a critical component in forming complex ecosystems that emerge from the interactions between biological, geochemical and geological processes[4]. In marine environments, methane cold seeps contribute to regional biodiversity by supporting specialized microbial and faunal communities adapted to harsh conditions[4–6]. In seep-impacted sediments, the anaerobic oxidation of methane (AOM) supports high fluxes of dissolved sulfide ($H_2S$)

[1]Department of Geosciences, UiT - The Arctic University of Norway, 9037 Tromsø, Norway. [2]Department of Earth and Environmental Sciences (DISAT), University of Milano Bicocca, Milan 20126, Italy. [3]Department of Biological Sciences and Centre for Deep Sea Research, University of Bergen, 5006 Bergen, Norway. [4]Norwegian Offshore Directorate, 9406 Harstad, Norway. [5]Centre for Environmental and Marine Studies (CESAM) & Biology Department, University of Aveiro, 3810-193 Aveiro, Portugal. [6]Dronningåsen 14, Stavanger NO-4032, Norway. [7]WHOI, Woods Hole Oceanographic Institution, Woods Hole, MA 02543, USA. [8]Minderoo-UWA Deep-Sea Research Centre, University of Western Australia, 6009 Perth, Australia. [9]Ocean Census, Begbroke Science Park, Oxfordshire OX5 1PF, UK. [10]National Oceanography Centre, European Way, Southampton SO14 3ZH, UK. [11]Reservoir Technology Department, Institute for Energy Technology, 2007 Kjeller, Norway. [12]LIttoral ENvironnement et Sociétés (LIENSs), La Rochelle Université, Bâtiment ILE, La, 17000 Rochelle, France. [13]Department of Geosciences, University of Oslo, 0371 Oslo, Norway. ✉e-mail: giuliana.panieri@uit.no

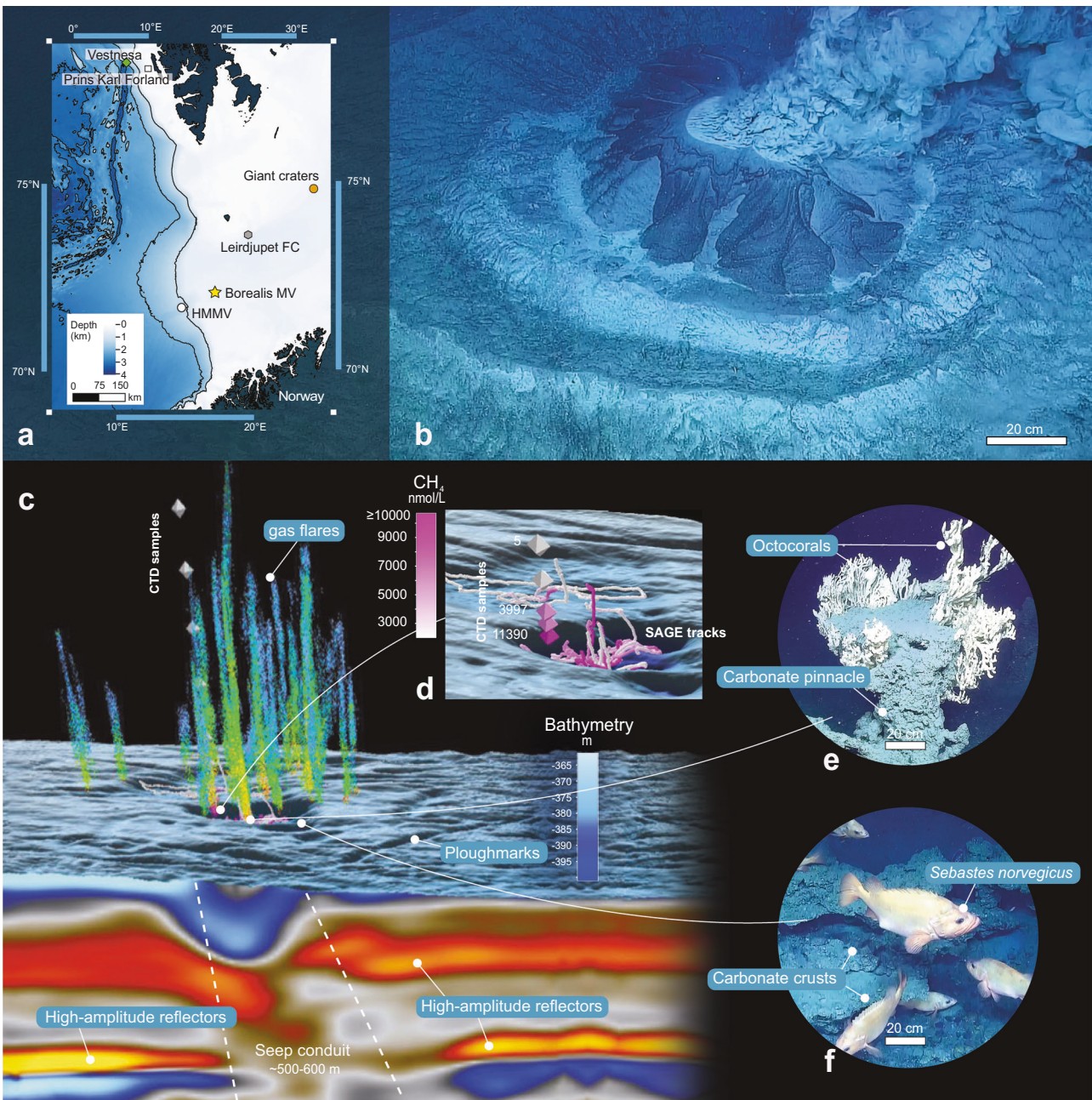

**Fig. 1 | Overview and observations of Borealis mud volcano. a** Map showing the location of Borealis mud volcano (yellow star) and other cold seeps in the area (for some of them such as: Prins Karl Forland in white square[71], from Leirdjupet Fault Complex (Leirdjupet FC) in grey hexagon[72], from Håkon Mosby Mud Volcano (HMMV) in white circle[73] and from Vestnesa Ridge (Vestnesa) in green diamond[7], the origin of emitting methane is shown in Fig. 3; Giant craters in orange circle are mentioned in the text. The map in Fig. 1a was created from the IBCAO bathymetric data[74]. **b** Active gryphon emitting warm fluid, methane and Neogene sediments. **c** Compiled observations, including seabed topography from high-resolution multibeam data (5 m grid cell), a seismic cross-section from 3D-seismic dataset SPE16M01 (the complete seismic section is available in Supplementary Fig. 1) and

the multi-beam echosounder data (320 kHz) tracing streams of gas bubbles (gas flares) in the water column with variations in the colours of flares representing the backscattering intensity of the reflected acoustic signals (red indicating the highest values and light blue the lowest). **d** Georeferenced detail of the confined depression (~0.14 km²) around the active gryphon showing the methane water concentration (represented with the same colour scale to maintain consistency) measured in CTD water samples (position of the water sampling indicated by the diamonds in the vertical line) and real-time SAGE measurements (data showed as ROV tracks) (Supplementary Data 3–6). **e** Georeferenced ROV images show a carbonate pinnacle colonised by Octocorallia (pinnacle high 120 cm) and **f** the red fish *Sebastes norvegicus* (~30 cm in length).

toward the seafloor, which is colonized by sulfur-oxidizing microbial mats and chemosymbiotic organisms, such as clams, mussels and tubeworms[6–8]. Methane seepage sites are often associated with widespread carbonate deposits that precipitate in situ due to the increased local alkalinity induced by AOM[9]. These deposits have a wide range of morphologies, from flat pavements to vertical pinnacle-like structures, and highly variable dimensions from a few

mm-sized concretions to beds several hundreds of meters in lateral extent and several meters in thickness[10]. Seep carbonates provide hard substrata for sessile organisms[4] and resources for other species in adjacent areas, contributing to a broader ecological network[6]. They are also a record of geological processes of methane oxidation spanning millions of years and have been used in high-resolution paleo-reconstructions[11–13].

Despite numerous observations of methane emissions from the seafloor in Arctic regions, only five mud volcanoes have been discovered in the Canadian Beaufort Sea (Western Arctic)[14], and three in Alaska[15], and so far, the Håkon Mosby Mud Volcano (HMMV) was the only known structure in Norwegian waters[16]. Mud volcanoes are surface manifestations of focused fluid flow in hydrocarbon-rich sedimentary basins along passive margins and at convergent plates characterized by high sedimentation rates[17]. Mud volcanoes collectively play a significant role in the atmospheric methane budget[18,19] by releasing an estimated 60 Tg $CH_4$ $yr^{-1}$ [19] sourced from several kilometres depth[17]. These estimates have large uncertainties since the total number of mud volcanoes worldwide and their temporal variability concerning methane emission rates are not confidently constrained, especially when considering undiscovered offshore mud volcanoes. Therefore, our current understanding of their potential contribution to atmospheric methane emissions and impact on climate, ocean chemistry and ecosystems is still limited. Likewise, the mechanisms triggering their activity and their role in promoting the survival of specialised fauna thriving at these sites remain debated[5,6].

Here, we present a comprehensive study of a recently discovered mud volcano in the Polar North Atlantic named Borealis mud volcano (MV) located in Outer Bjørnøyrenna (Bear Island Trough, 72° 26.304´N, 17° 40.626´E, -390 m water depth) in the Barents Sea (Fig. 1). This discovery marks the second mud volcano of this kind identified in the region, among numerous methane seeps previously detected[20], thus opening a *new* chapter in our understanding of Arctic geology and related fields of research. Borealis MV is characterized by a cluster of craters from -70 m to -400 m in width within a major depression of 500–600 m in diameter and an active gryphon (~ 7 m in diameter and 2 meters high) expelling warm fluids, gas and oil. Our observations also show that Borealis MV acts as a sanctuary for fauna vulnerable to anthropogenic perturbations, specifically seabed trawling, which continues to have a significant impact on benthic ecosystems in the region[21,22].

## Results

### Subseafloor, seafloor and water column insights

Borealis MV is located in the eastern parts of the Sørvestsnaget Basin, where a thick sequence of Cretaceous and Cenozoic sedimentary rocks is covered by a wedge of Pliocene to Pleistocene sediments[23,24]. In this area, industry 3D-seismic data revealed the presence of a 500–600 m wide seismic chimney that extends from a depth of 300 m below seafloor (m bsf) and terminates at the seafloor into a complex network of crater-like depressions (Supplementary Fig. 1). Shallow bright amplitude acoustic anomalies observed in the Quaternary sediments are interpreted as local free gas accumulations within a vertical chaotic zone. The roots of the acoustic chimney are located at the base of the Quaternary/URU (Upper Regional Unconformity) boundary (Fig. 1).

The detection of the chimney structure steered our hydroacoustic observations, which subsequently revealed persistent, concentrated and active gas flares originating from the Borealis MV. Within a confined area (-0.14 km²) around an active gryphon, we have identified a total of 26 gas flares (Fig. 1). These individual flares exhibit a typical height of 160 m on average, potentially ascending up to 355 m into the water column. In some instances, flares even approach the sea surface (Supplementary Fig. 2). CTD water samples collected from one of these plumes showed the presence of methane with concentrations that remain relatively high throughout the water column and preserve notable levels, up to 11.390 nmol $L^{-1}$ close to the seafloor (Supplementary Data 2). This suggests an intense release of hydrocarbon-charged fluids. In comparison, background ambient values of methane concentrations are around 0.9 nmol $L^{-1}$, making the observed values approximately 100 times higher than the ambient conditions. Additionally, the spatial scale survey conducted with the sensor SAGE (Sensor for Aqueous Gases in the Environment) across the main craters

of the mud volcano revealed that the high methane emissions from Borealis significantly affect all the surrounding areas, demonstrating the extensive impact of the active hydrocarbon emissions. SAGE, which uses a deep-sea membrane inlet to extract dissolved gas from seawater, revealed methane background levels (212 nmol $L^{-1}$) to be well above the typical background ambient levels. The highest values of methane (levels ≥10.000 nmol $L^{-1}$; Fig. 1 and Supplementary Fig. 3; Supplementary Data 3–6), were measured within the southern craters, where several flares and the gryphon were observed.

The subsequent ROV seafloor observations confirmed the persistent and ongoing methane venting from four main craters, within a major depression (Fig. 1). The two northernmost craters display extensive carbonate deposits exposed at the seafloor as stacked slabs forming tens of meters wide large pavements. The thickness of single slabs is in the order of tens of cm, but the total thickness of the deposits can reach -5 m along the steep wall of the craters. While carbonate crusts have been observed in other cold seeps in the Barents Sea[10,11,25], the carbonates at Borealis are remarkably larger and thick (Fig. 2), emphasizing the exceptional nature of this site. The gryphon, on the southern flank of the deepest crater of the Borealis MV, has a central conduit that serves as the epicentre from which warm fluids and sediments are actively expelled, as observed from gas bubbles and sediment flows extending over the seafloor.

### Origin of fluids and expelled mud

We collected the expelled unconsolidated sediments from the gryphon using a push core manipulated by the ROV (Supplementary Fig. 3). The sediments consist of medium-sized sand (grains from 0.25 to 0.5 mm), with minor clayey content (particles <0.002 mm), and a few small sedimentary rock fragments and coal pebbles (typically <2 cm)[26]. The expelled sediment contains planktonic foraminiferal fauna correlated with the micropaleontological zonation for the Neogene on the Vøring Plateau[27] while the benthic foraminiferal fauna correlated with the one for the Cenozoic of the North Sea[28]. Nearly all the individuals are extinct species typically associated with Pleistocene deposits on the Norwegian Shelf from ca[29] 700–1000 m bsf. The lack of benthic foraminifera typical for Holocene and recent sediments on the Norwegian continental shelf, including *Trifarina angulosa* and *Uvigerina peregrina*[30–32], and the almost complete lack of warm water-dwelling planktonic foraminifera indicate that no Holocene sediments are present in the erupted material.

Geochemistry performed on the sediment from the push core indicates that the fluid emitted from Borealis gryphon has lower levels of organic carbon and nitrogen compared to the average southwestern Barents Sea surface sediments (i.e. TOC > 0.5 %; TN > 0.05 %)[33] with values of 0.33 wt.% and 0.02 wt.%, respectively. The isotopic composition of bulk decarbonated material ($\delta^{13}C = -27.11$ ‰ and $\delta^{15}N_d = 4.58$ ‰), associated with the high C/N ratio (20.2), are consistent with a high contribution of ancient organic sources that underwent prolonged microbial degradation in the deep subsurface[34,35]. Organic biomarkers revealed the presence of detectable amounts of crude oil associated with immature bitumen showing virtually no biodegradation (Supplementary Fig. 4). The source rock appears to be marine shale/lacustrine deposits of Jurassic age or younger. Recent studies reported the widespread and extensive methane and oil release from geological reservoirs to the Arctic Ocean[20] and, although evident oil seepage or oil slicks were not observed at Borealis MV, this site can be considered as the surface expression of a deeper seated active petroleum system.

The 30-minute ROV dive over the gryphon revealed frequent (every 5–10 minutes) eruptions of warm water-mud-gas mixture that rises along the seep conduit below (Fig. 1).

The gas composition of the bubbles escaping from the sediment is primarily methane, characterized by a typical microbial isotopic signature[36] (Fig. 3). Additionally, elevated fractions of ethane and propane, measured in a gravity core taken from outside the craters,

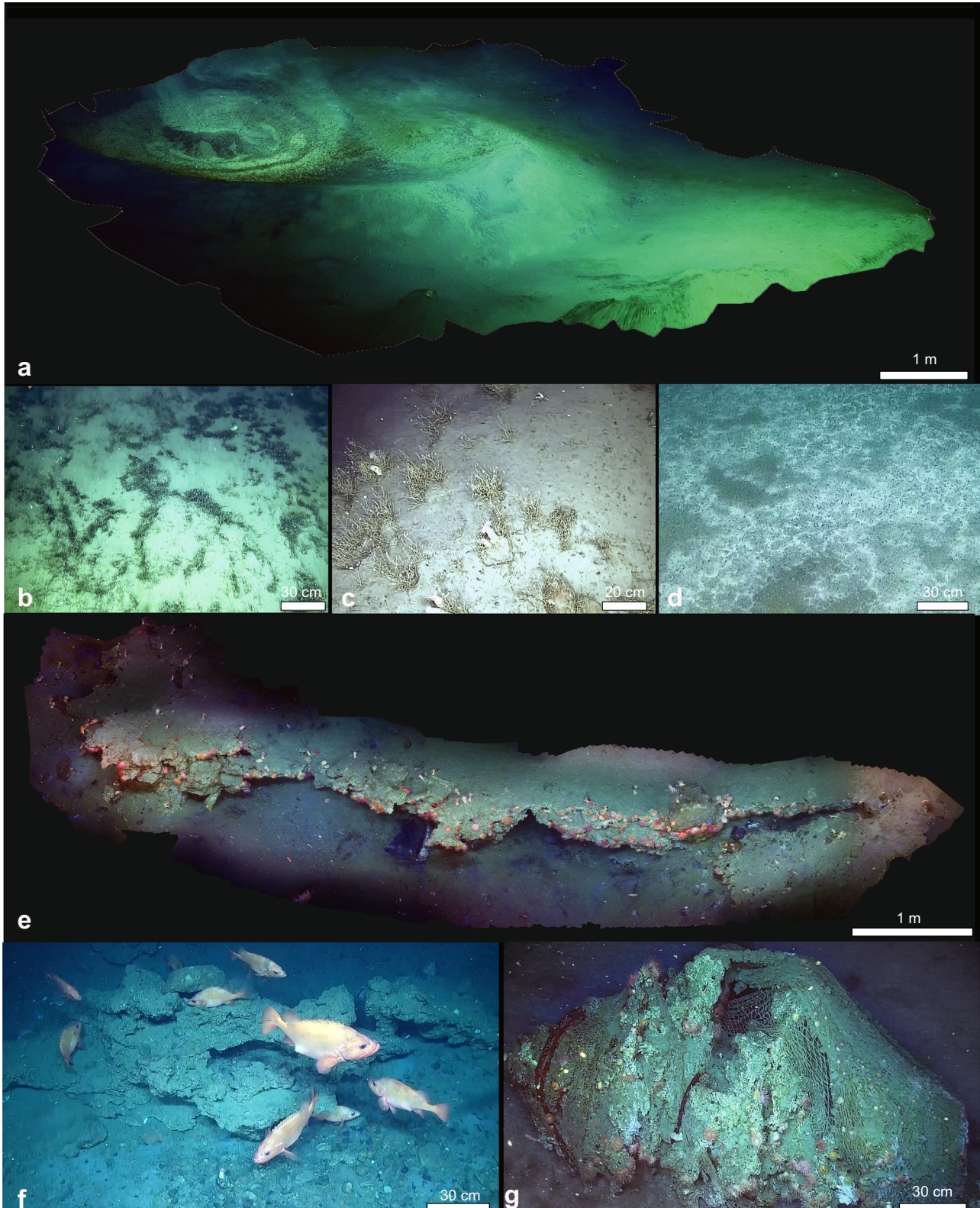

**Fig. 2 | Detailed ROV observations of Borealis mud volcano. a** mosaic of the Borealis MV; **b** tubeworm aggregations (*Oligobrachia* sp.), and **c** the dense colonies of hydrozoans (*Tubularia* sp.) located in the slope area of the volcano crater; **d** extensive microbial mat; **e** a mosaic of the carbonate structures colonized by sessile fauna and used by various fish species, such as **f** the redfish *Sebastes norvegicus* as breading grounds and refuge areas; **g** lost fishing gear stuck on the carbonates structures and colonized by the typical sessile fauna observed in the region.

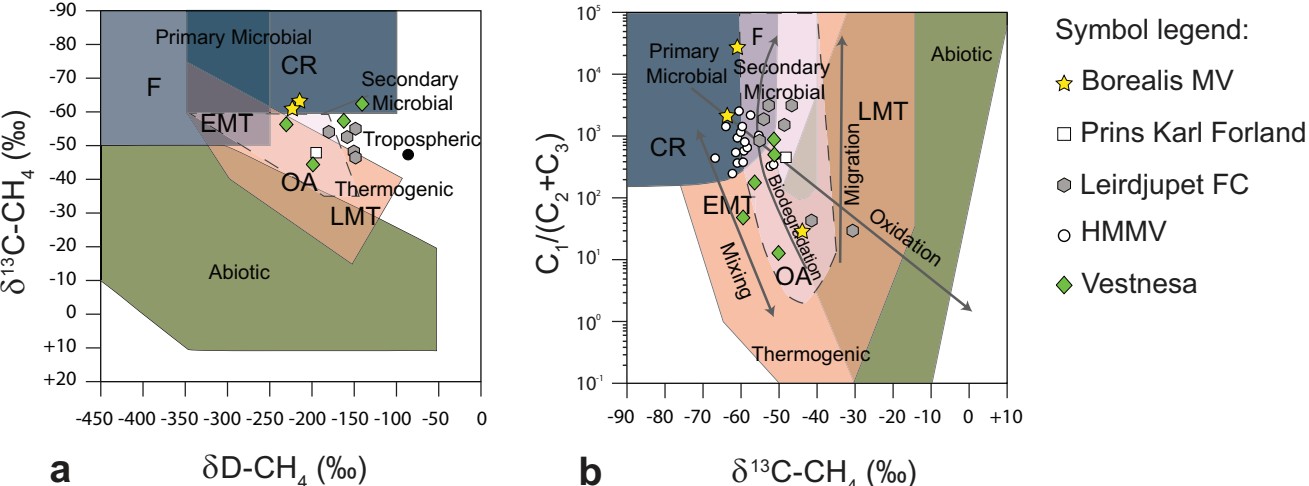

**Fig. 3 | Geochemistry of the gas emitted from the Borealis mud volcano.**
**a** Stable carbon ($\delta^{13}C$) and hydrogen ($\delta D$) isotope composition of methane from headspace gas analysis. Sample data from Borealis MV are reported in yellow starts, and for comparison, other high-latitudes cold seeps (location in Fig. 1) are reported: from Prins Karl Forland in white square[71], from Leirdjupet Fault Complex (Leirdjupet FC) in grey hexagon[72], from Håkon Mosby Mud Volcano (HMMV) in

white circle[73] and from Vestnesa Ridge (Vestnesa) in green diamond[7]. Genetic fields of hydrocarbons (CR–$CO_2$ reduction, F–methyl-type fermentation, EMT–early mature thermogenic gas, OA–oil-associated thermogenic gas, LMT–late mature thermogenic gas) after[36]. **b** Plot of $\delta^{13}C$-$CH_4$ versus the composition of light hydrocarbon components ($C_1/(C_2 + C_3)$ ratio). Grey arrows indicate the main processes affecting the isotopic and molecular compositions of gases.

suggest the presence of a deeper thermogenic component. Such spatial and temporal variations in subsurface fluid migration are typical at mud volcanoes and other cold seep environments and could account for the observed variability in gas compositions. Within the craters, it is possible that the microbial gas masks the presence of the thermogenic component.

The temperature of the discharged fluid measured above the active gryphon by the CTD installed on the ROV was 11 °C, and that of the T-probe was 11.42 °C. This temperature is substantially higher than the reference ca. 4 °C measurements away from Borealis.

## A unique oasis for faunal communities

Borealis MV hosts a diverse array of faunal communities, including seep-associated and background fauna, together with species that have commercial value. Within the MV´s craters, we observed in the ROV video dense and extensive patches of microbial mats and sibo-glinid tubeworm aggregations (*Oligobrachia* sp.; Fig. 2) akin to those documented at other Arctic cold seeps[8]. Microbial mats and tube-worms seem to be the main characteristic of high-latitudes compared to lower-latitudes cold seeps, which host clams, mussels and vesti-mentiferan tubeworms[4]. The microbial mats, extending several square meters, form the foundation for a variety of microorganisms, including a multitude of foraminifera species. Data obtained from environmental DNA (eDNA) analysis revealed the presence of distinct species of hard-shelled foraminifera (*Epistominella*. sp., *Reophax dentaliniformis*, *Stainforthia* sp.), monothalamids and soft-shelled foraminifera nestled within these microbial ecosystems. Although, until now, no endemic foraminifera species have been documented within cold seeps[37], our preliminary eDNA data from Borealis MV suggest the possibility of previously unidentified species potentially unique to these types of environments.

The megafauna diversity at Borealis MV, as observed from ROV imagery, is relatively low, and appears to be influenced by several environmental factors, including elevated methane concentrations, extensive carbonate crusts on the seafloor and the suspended sedi-ment particles emitted by the gryphon that further contribute to the challenging conditions that may be impacting the larger faunal assemblages. The megafauna is dominated by clusters of anemones and serpulids anchored to carbonate substrates and hydrozoan

colonies (*Tubularia* sp.) that flourish on the crater slopes, while the presence of cladorhizid sponges and sea stars is more sporadic. In addition, our sample collections from various habitats within the Borealis MV have yielded taxa not detectable via ROV imagery, such as annelids, amphipods, gastropods, polyplacophorans, nemerteans (*Nipponemertes* spp.), and ophiuroids (Supplementary Data 1, Sup-plementary Fig. 5). Morphological and molecular analyses of these samples are currently underway to further elucidate the composition of these faunal communities. The observed scarcity of megafauna taxa suggests that the immediate environmental conditions sur-rounding the Borealis MV may be inhospitable for a broader range of organisms. Remarkably high methane concentrations (11.390 nmol L$^{-1}$ near the seafloor) or toxic sulfide levels are factors known to sig-nificantly influence invertebrate community structure at seep sites[38]. These chemically-enriched habitats create a gradient of extreme environmental conditions that can be detrimental to many forms of marine life, often resulting in reduced biodiversity. Additionally, the high volume of suspended sediment particles emitted by the gryphon may impact the diversity and density of filter- and suspension-feeding organisms, potentially by clogging their feeding apparatus[39]. Never-theless, some taxa appear resilient to the presence of suspended sediment. Hydrozoans, for example, are abundant in the carbonate area covered with sediment particles, similar to what was also observed on the Koryak slope at upper bathyal depths (~660 m)[40].

While some parts of the seafloor within Borealis MV seem to be inhospitable for many organisms, the extensive carbonates provide additional habitat and suitable hard substratum for epifauna including dense aggregations of several species of anemones, serpulids, demosponges, nudibranchs, and octocoral colonies (*Primnoa rese-daeformis*). *P. resedaeformis* was exclusively found in the jagged car-bonate area (Fig. 2), characterized by minimal or absent sediment deposition from the gryphon plumes (Supplementary Data 8 and Fig. 4). As observed earlier at other offshore seepage sites[4,41–43], car-bonate deposits represent oases for numerous sessile organisms. Moreover, the carbonate structures may offer both shelter and feeding opportunities, thereby playing a role in sustaining the local fish populations. We observed large schools of saithe (*Pollachius virens*) and various demersal species such as spotted wolffish (*Anarhichas minor*), cod (*Gadus morhua*), four-bearded rockling (*Enchelyopus*

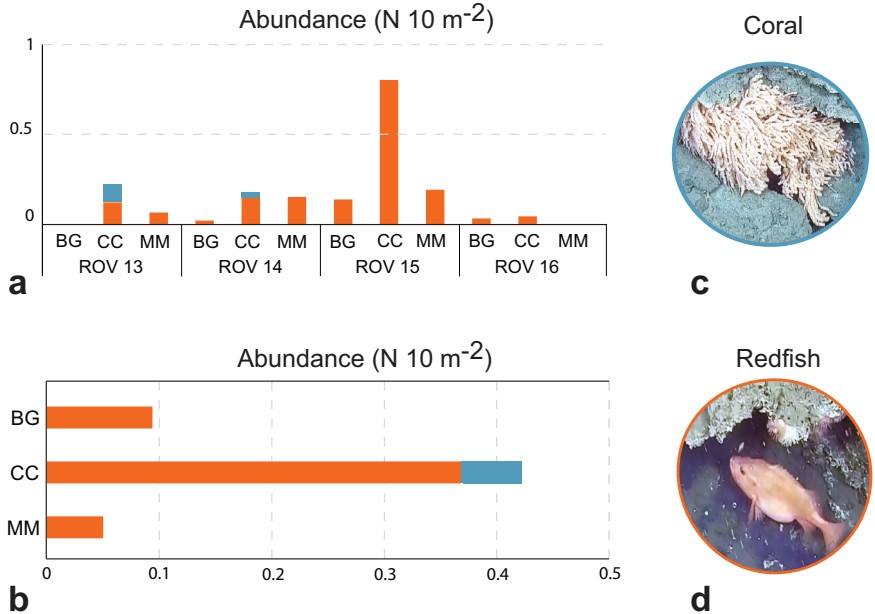

**Fig. 4 | Abundance of the main vulnerable megafauna species observed at Borealis mud volcano.** The substrate is differentiated in carbonate crusts (CC), microbial mats (MM) and background (BG) (details for the definition of the substrate in the Methods section), while the megafauna is distinct in redfish (all encountered species of redfish such as *Sebastes norvegicus, S. viviparus* and *S. mentella*; displayed with orange bars) and Corals (including all species of octocorals, such as *Primnoa resedaeformis* and others classified at lowest possible taxa level through videos; displayed with light blue bars). **a** Abundance of redfish and corals in the three different environments during each ROV dive. **b** Total abundance of redfish and corals in all the ROV dives meant to display their overall trend in the entire Borealis site. The megafauna abundance is expressed as the number of occurrences per 10 square meters ($10^{-2}$ m). Data reported in Supplementary Data 8. **c** Example of an Octocoral (screenshot from ROV Dive 15; Supplementary Data 1). **d** Example of a redfish (screenshot from ROV Dive 15; Supplementary Data 1).

*cimbrius*), and several species of redfish (*Sebastes* spp.) clustering around the jagged carbonate formations (Fig. 2).

There are three species of *Sebastes* common in the Barents Sea (*S. norvegicus, S. mentella* and *S. vivlparus*)[44,45], and it was not possible to separate these three with certainty using the ROV footage. However, based on the presence/absence of a pronounced protrusion on the lower lip (beak), proportional size of the eye and presence/absence of a dark patch on the gill covers, we assume that *S. norvegicus* was the most commonly observed, although all three species appear to be present. *S. norvegicus* is listed as endangered on the Norwegian Red List for Species and is subject to a fishing moratorium[45,46]. The redfish were particularly numerous and clustered close to the carbonate structures. In the ROV videos, we observed several redfish individuals with clearly distended abdomens (Fig. 2). The complex three-dimensional structures and the irregular morphology of the carbonates provide protective shelters, and the site's elevated temperatures may enhance reproductive success by accelerating egg development[47].

During our expedition, we encountered lost fishing gear (Fig. 2) from bottom trawling snagged on the jagged carbonate rock around the perimeter of the craters. These were colonised by sessile fauna (e.g. anemones and hydrozoans), suggesting that the gear was lost several years ago. The current cessation of fishing activities in the Borealis area is confirmed by the Vessel Monitoring System (VMS)[48] and the Automatic Identification System (AIS) data from the OSPAR/ICES database[49] and the Global Fishing Watch database[50] on bottom fishing intensity for the area. This information, in conjunction with the presence of red-listed species *S. norvegicus* and taxa indicative of Vulnerable Marine Ecosystems (VMEs) as defined by FAO/ICES[51], such as the octocorals, suggests that the Borealis MV is a *de facto* sanctuary for these endangered species. The absence of bottom trawling and the natural habitat's protective qualities offer a refuge where these species can thrive despite significant seafloor impacts from fishing in the surrounding Barents Sea[21,22].

## Discussion
### Glacial history and genesis of Borealis MV
The Borealis MV is the second mud volcano ever identified in Norwegian waters since the discovery of HMMV[16]. The latter is located 110 km southwest of Borealis at ~1260 m water depth. Previous studies suggest that its activity started ~330 ka before present, when fluids were expelled from the periglacial units loaded by a 3 km thick glacial deposit[52]. For Borealis MV, we propose a different genesis and a different age. During the Last Glacial Maximum (LGM) ~23 ka before present, the expansion of the Eurasian Ice Sheet (EIS) dominated the Barents Sea landscape[14]. Well-defined ploughmarks surrounding Borealis MV give evidence of seafloor erosion following the LGM. However, the lack of evidence of such ploughmarks intersecting the mud volcano craters indicates that the latter must have formed after the deglaciation phase (Supplementary Fig. 6). One possible formation scenario is consistent with those depicted for the large Troll pockmark field[13] or the nearby giant craters in Bjørnøyrenna[53], that we termed "deglaciation trigger model". As ice retreated, warming temperatures and a decrease in pressure destabilised methane hydrates once trapped within the sediments. The dissociation of these gas hydrates suddenly liberated large amounts of methane, which triggered the formation of craters on the seafloor. The observed widespread carbonates contributed to cementing the sediments, reducing the porosity/permeability and ultimately acting as a buffer layer for the rising methane-rich fluids. Those carbonates formed barriers that periodically inhibit the upward migration of hydrocarbons, allowing gas accumulation underneath the carbonate pavements[42,54] and forcing gas to find alternative pathways to the seafloor. This deflection of the fluid flow might have contributed to the genesis of the four observed craters in a sequential and ongoing geological process in the Borealis MV system, possibly by pressure buildup and abrupt release via explosive events. Diffuse, long-lasting gas hydrate dissociation is still currently ongoing at northern latitudes, as observed in numerous sites

in the Barents and Norwegian seas[53,55]. This underscores the enduring influence of deglaciation on contemporary geological and environmental processes at high latitudes.

The temperature anomaly measured at Borealis clearly indicates that the mud volcano plumbing system is connected to deeper and warmer strata from which fluids rapidly migrate towards the surface. This observation is also very relevant in explaining potential triggers for the eruption model of Borealis MV. In addition to the "deglaciation trigger model" proposed above, we also suggest a "hot fluids surge" scenario to explain the formation of the craters. Similarly to what has been observed at numerous mud volcanoes in Lake Baikal[56,57], we suggest that 1) batches of upwelling warmer fluids periodically dissociated gas hydrates deposits at shallower depth, resulting in 2) increased volume and pressure that entrained the sediments, then causing 3) multiple surface eruptions.

Since significant methane content was measured close to the sea surface, we may also speculate ongoing atmospheric emissions under favorable oceanic conditions. However, while the present contribution of atmospheric carbon from marine geological sources is deemed relatively minor, considerable uncertainty persists on future emission projections on a warming planet, underscoring the importance of measuring methane emissions from cold seeps like Borealis MV.

### A natural sanctuary for threatened Arctic species

The Borealis MV may also play a complex role in supporting local marine life. Many fish species observed at Borealis MV have been identified in natural (e.g. *Lophelia* reefs)[58] or artificial reef-like structures on the Norwegian shelf and slope[59] and may use these habitats over the long term or transiently during migrations as part of their life cycle. There have been several observations of animals in the deep sea aggregating and spawning around seeps and vents[60,61]. The direct causation of such aggregations is not known for sure; however, several explanations have been offered. First, deep-sea organisms often exist at low population densities, so aggregation around prominent topographic features can be advantageous for reproduction[62]. Second, the conditions around seeps provide enhanced local productivity[4]. These offer more feeding opportunities for both adult and juvenile stages of organisms. Third, the elevated temperatures around seepage sites may also create optimal conditions for reproductive processes, leading to more effective breeding cycles. Elevated temperatures are likely to enhance the development times of eggs and larvae of organisms, with Drazen et al. (2003)[60] estimating an increase in temperatures of 1.5 °C at the Gorda Ridge in the Pacific could reduce egg incubation time by 10% in benthic fish and octopus assuming a coenzyme Q10 of 2. Several individuals of redfish observed in the ROV videos had clearly distended abdomens (Fig. 2), and considering that these observations were made during the peak spawning season for redfish (early May[63,64]), they may be spawning in this area. The carbonate structures may even serve as nursery areas for redfish larvae, with their irregular morphology offering shelter and the site's elevated temperatures potentially enhancing reproductive success by accelerating egg and larval development[47]. The taxa observed at Borealis, such as the redfish, have a different reproductive ecology than the species that have been recorded previously to use vents or seeps for the incubation of eggs. Unlike those species, redfish undergo direct development, releasing free-swimming larvae that might not be as affected by the increased temperature compared to eggs incubated directly on the seafloor. Still, the environment around the Borealis MV may provide elevated food and temperature and an obvious "reef-like" effect, aggregating fish that take advantage of shelter and protection from bottom trawling. Given the conservation status of the fish in the Barents Sea, with some species threatened due to overexploitation, even a relatively small location like the *Borealis* MV and associated carbonates could be significant as a population refuge.

Furthermore, preserving ecosystems like *Borealis* MV is crucial for biodiversity conservation and a comprehensive understanding of the intricate interactions between geology, geochemistry and biology in marine environments. The Arctic seafloor has become a vital asset, playing an important role in oil and gas exploitation activities and the emerging deep-sea mining industry. The responsible management of marine mineral and biological resources is paramount for sustainable development and environmental stewardship in the Arctic region. In the longer term, Norway has committed to the 30×30 target (protecting 30% of land and sea by 2030) for spatial conservation measures of representative marine ecosystems, including in the deep sea[65]. Protecting large areas of the deep-sea floor along the Norwegian margin may result in seep refugia acting as source populations for wider recolonization and restoration of benthic biological communities.

## Methods

### Multibeam echosounder Water Column Data

Borealis MV's morphometric data were obtained from a 3 × 3 m resolution morphobathymetric map generated on board and obtained by processing multibeam echosounder data (i.e., depth measurements and backscatter) acquired using the hull-mounted Kongsberg MBES EM710, from which water column data were also recorded, documenting evident flares. We received permission from NOD, The Norwegian Offshore Directorate, to collect data from the Borealis study site.

Gas seep mapping at the study site was done using the QPS FMMidwater software. Gas seeps were detected as gas flares in the water column data caused by backscatter from the gas bubble streams. The acquired data from EM302 multibeam systems in *.all and *.wcd file formats were converted with FMMidwater to the generic water column format (*.gwc). *.gwc files were visualised in fan view and stacked view. The fan view allows for narrowing the opening beams to select individual flares in the stacked view and export them in a sd file for visualisation in Fledermaus. Only one flare was kept when multiple flares showed the same source in Fledermaus. Locations of individual gas flares were retrieved from the lowest point of the flare in the sd file. This identification is enabled by the significant differences in velocity and density between chains of gas bubbles and the water column, leading to pronounced contrasts evident in the acoustic signals. Flare locations were plotted on the maps and used during ROV-flying during the ROV dives to locate streams of gas bubbles.

### CTD

Temperature and salinity at specific depths in the water were obtained from a CTD (Conductivity, Temperature, Depth) mounted on a rosette, which was lowered in the water column from the hull of the vessel. Twelve 10-litre Niskin bottles were also mounted on the rosette, which we closed at specific depths to estimate dissolved methane.

### ROV and video analyses

The Aurora work-class ROV is a 6000 m depth-rated ROV with a tethered management system (TMS) called Borealis and provides unique opportunities for science and filmmaking. Four dives (ROV13, ROV14, ROV15, and ROV16) were conducted along exploratory track lines crossing four craters (for a total of 14 hours and 4 minutes of diving time). The SfM (Structure from Motion), photogrammetry technique[37] was applied on selected ROV video frames to develop 2D photo-realistic orthomosaics of representative physical habitats and sedimentary facies (Fig. 2).

All the collected videos were analyzed for substrate characterization and the recording and counting of the main vulnerable megafauna observed in the surveyed areas. To ensure the reliability of our seafloor classification, the substrate was differentiated into carbonate crusts (CC), Microbial Mats (MM), which are the more typical and

representative seafloor types for a cold seep and all other substrates not clearly associated with seeping phenomena. All detected megafauna occurrences were considered variables and classified as Redfish and Coral. The Redfish category comprises all encountered species of redfish such as redfish (*Sebastes norvegicus, S. viviparus* and *S. mentella*) (denoted as RedF). The Coral category includes all species of octocorals (such as *Primnoa resedaeformis* and others classified at the lowest possible taxa level through videos). Each ROV dive surveyed different areas, with some level of overlapping between ROV tracklines crossing the Borealis MV main seeping sites (Supplementary Fig. 7). The surveyed area covered by each ROV dive was then estimated using a GIS-based tool and applying a $1\,m^{-2}$ buffer area surrounding each ROV position to remain conservative in estimating the observed area by focusing on areas well-framed by videos and easy to interpret. Because of the differences in the extent of the areas surveyed by each video, the abundance of the detected megafauna (Redfish and Coral) is expressed as number (N) per $10\,m^{-2}$ in Supplementary Data 8. Data are shown in Fig. 4.

### Fluid analyses

**Temperature measurements**. The temperature (T) of the fluid emanating directly from the crater of the Borealis MV was measured using the CTD sensor and a temperature probe, the ISD400 Depth and Temperature Sensor from Impact Subsea (precision level of ±0.01% °C), both mounted on ROV Aurora.

**Methane measurements**. Water samples collected from the CTD Niskin bottles were transferred into 120 mL glass bottles containing 5 mL of 1 M NaOH. The samples were stored in the dark at 4 °C until analyses with a Gas Chromatographer - FID (ThermoScientific Trace 1310). Before the analyses, we created a 5 mL headspace and let the samples equilibrate overnight. Measurement analyses for each sample collected at different water depths are reported in Supplementary Data 2.

The dissolved methane was also continuously measured during the ROV dives using SAGE (Sensor for Aqueous Gases in the Environment), a dissolved methane instrument designed and built in the Chemical Sensors Laboratory at the Woods Hole Oceanographic Institution (WHOI)[38,39]. SAGE has a detection range of 5–10,000 ppm $CH_4$. SAGE uses a deep-sea membrane inlet to extract dissolved gas from seawater. Inside the instrument, the extracted gas fills a hollow core optical fibre. Laser spectroscopy measures the methane inside the optical fibre by coupling the light from a laser to the fibre. The dissolved methane data are reported in Supplementary Data 3.

### Geochemistry

**Sediment geochemistry**. We prepared 0.3 g of dry sediment from a surface sample collected at the seeping spot to measure its organic carbon, nitrogen content, and isotopic composition ($\delta^{13}C$, $\delta^{15}N$). The carbonate material was removed by acid addition using 6 N HCl. Analyses on decarbonated material were conducted at the SIL (UiT) using a Thermo-Fisher MAT253 Isotope Ratio Mass Spectrometer (IRMS) coupled to a Flash HT Plus Elemental Analyzer. The $\delta^{13}C$ and $\delta^{15}N_d$ values were determined and normalised to Vienna Pee Dee Belemnite (VPDB) ($\delta^{13}C$) and Air-$N_2$ ($\delta^{15}N$) using 3 in-house urea and peptide calibrators. The analysis of soil control samples yielded a measurement repeatability (1 s; $n = 6$) of 0.01‰ on $\delta^{13}C$ and 0.15‰ on $\delta^{15}N$. The C/N atomic ratio was calculated using the atomic mass weighted ratio of TOC (Total Organic Carbon) and TN (Total Nitrogen) as $C/N = (TOC/12.011)/(TN/14.007)$.

**Oil geochemistry**. Sediment samples for oil geochemistry were collected from sediment slices, wrapped in aluminium foil and stored at −20 °C for freeze-drying. All oil preparation and analysis procedures followed NIGOGA (Norwegian Industry Guide to Organic Geochemical Analysis), 4th Edition, and were conducted at Applied Petroleum Technology (APT, Oslo). Extractions were performed with a Soxtec Tecator unit and dried before deasphaltering. A small amount of dichloromethane (3 times the amount of EOM, Extractable Organic Matter) is added. Pentane is added in excess (40 times the volume of EOM/oil and dichloromethane). The solution is stored for at least 12 hours in a dark place before centrifugation and the weight of the removal of asphaltenes. Quantifying saturates, aromatics A and polar (NSO-fraction) were done using two HPLC pumps, a sample injector, a sample collector and two packed columns. The pre-column is filled with Kieselgel 100 and heated at 600 °C for 2 hours to deactivate it. The main column, a LiChroprep Si60 column, is heated at 120 °C for 2 hours with a helium flow to make it water-free. Approximately 30 mg of deasphalted oil or EOM diluted in 1 ml hexane is injected into a sample loop. The solvents used are hexane and dichloromethane. The stable carbon isotope composition of the different fractions was measured on a Delta V Plus Isotope Ratio Mass Spectrometer (IRMS) (Thermo Fisher Scientific) via Conflo IV. A standard (NGS NSO-1, topped oil) is analysed for each $12^{th}$ sample. The $\delta^{13}C$ value obtained for this standard is −28.6‰ vPDB. The variation in the isotopic values for NSO-1 by repeated analysis over one year is ± 0.09‰. Age-specific biomarkers were measured via GC-MS/MS using a Thermo Scientific TSQ Quantum instrument. The column used is a 60 m CP-Sil-5 CB-MS with an i.d. of 0.25 mm and a film thickness of 0.25 μm. d4-27ααR was used as an internal standard.

**Gas geochemistry**. Two samples of seep gas were collected during the AKMA3[66] and Extreme24 expeditions using a bubble catcher and stored in steel flasks. One headspace gas sample was also obtained from a gravity core: a slice of sediment (-200 mL) was sampled from the bottom of the core and placed into an IsoJar™ paint can (Isotech Laboratories and Humble Instruments, USA), to which we added 0.5 mL of 10% benzalkonium chloride as antimicrobial agent and tap water. All the gas samples were stored at 4 °C. Aliquots of the samples were injected into an Agilent 7890 RGA GC equipped with Molsieve and Poraplot Q columns, a flame ionisation detector (FID) and 2 thermal conductivity detector (TCD). Hydrocarbons were measured by FID. The carbon isotopic composition of methane was determined via GC-C-IRMS. Aliquots were sampled with Triplus RSH autosampler and analysed on a Trace 1310 gas chromatograph (Thermo Fisher Scientific), equipped with a Poraplot Q column and PTV (Programable Temperature Vaporizing) injector. The GC is interfaced via GC-Isolink II and Conflo IV to Delta V Isotope Ratio Mass Spectrometer (IRMS) (Thermo Fisher Scientific). Repeated analyses of standards indicate that the measurement repeatability for $\delta^{13}C$ is better than 1 ‰ vPDB (2 s). The hydrogen isotopic composition of methane was determined by a GC-H-IRMS system. Aliquots were sampled with a Triplus RSH autosampler and analysed on a Trace 1310 gas chromatograph (Thermo Fisher Scientific) equipped with a Poraplot Q column and PTV (Programmable Temperature Vaporizing) injector. The GC is interfaced via GC-Isolink II and Conflo IV to Delta V Isotope Ratio Mass Spectrometer (IRMS) (Thermo Fisher Scientific). Repeated analyses of standards indicate that the measurement repeatability for δD is better than 10 ‰ vSMOW (2 s).

### Biological samples and observations

**Megafauna**. The description of the fauna present and distribution across the main micro-habitats of the Borealis MV was obtained through the cataloguing and annotation of all the ROV Aurora imagery collected during the expedition. Additionally, experts collected and identified physical samples of both specimens and sediments to support the correct identification of the specimens observed in the videos. A complete list of the samples observed is reported in Supplementary Data 1. Fauna attached to collected

carbonates was carefully removed with forceps and a scalpel, and the carbonates were thereafter rinsed with filtered seawater to collect small-sized specimens. Sediment samples were carefully sieved through a stack of sieves between 2 mm–0.5 mm to extract the megafauna and fixed on absolute ethanol to enable downstream DNA extraction. It is important to note that not all samples were analyzed by experts during this initial phase since a more detailed and comprehensive investigation of these samples is planned to further enhance the understanding of the Borealis ecosystem. Fauna samples will be deposited in the University Museum of Bergen collections.

**Foraminifera (eDNA).** The foraminiferal species list (Supplementary Data 7) was inferred from two sediment samples collected from a microbial mat (Supplementary Fig. 4). Briefly, after the DNA extraction using the DNeasy PowerLyzer PowerSoil kit (Qiagen, Germany), the specific foraminifera 37 f hypervariable gene of 18S rRNA gene was amplified with foraminifera specific primers 14F1 (14F1 (5′-AAGGG-CACCACAAGAACGC-3′) and s15 (5′- CCACCTATCACAYAATCATG -3′) primers. Per sample, a different combination of tagged primer was used, and three PCR replicates were performed. The PCR products were then verified on agarose gel, pooled and added to a library. The sequencing library was prepared using TruSeq® DNA PCR-Free Library Preparation Kit (Illumina) and quantified by qPCR using Kapa Library Quantification Kit for Illumina Platforms (Kapa Biosystems). The library was paired-end sequenced on a MiSeq instrument using the kit v2 (300 cycles). The raw data was demultiplexed with a DTD module from SLIM[67]. We used then DADA2[68] to generate Amplicon Sequencing Variants (ASV). Only ASVs containing foraminiferal "GACAG"[69] pattern and with at least 100 reads were retained. Those ASVs were finally taxonomically assigned using VSEARCH[70] with 90% min. similarity against a foraminifera database.

**Biostratigraphy and sedimentology**
The top 3 cm of a sediment core collected from the rim of the active emitting fluid gryphon (AKMA 3 ROV16 PusC C5-M05, Supplementary Fig. 5) was analysed for biostratigraphy to date the formation from which the emitting sediments originate and correlate them with known geological time periods. The undried, unconsolidated sample was soaked in water, wet sieved and dried. Then the dry sample material was fractionated. The fraction of 0.1–0.5 mm was gravity-separated in heavy liquid. The air in the foraminiferal chambers caused them to float up, and the tests could be collected for identification. The fraction less than 0.06 mm (silt and clay) was washed down the sink. The other fractions were used for a simplified grain distribution analysis. The fractions larger than 0.5 mm and less than 0.1 mm were also examined to investigate whether any important foraminifera were left in these fractions. All foraminifera and some other microfossil of importance from all fractions were investigated and recorded (more than 300 individuals) using a stereomicroscope. For planktonic foraminifera we used the zonation of Spiegler and Jansen (1989)[27] for the Neogene on the Vøring Plateau while the benthic foraminiferal fauna can be correlated with the micropaleontological zonation of King (1989)[28] for the Cenozoic of the North Sea. Nearly all the forms are extant species typically associated with Pleistocene deposits on the Norwegian Shelf. The sediment size composition was described following the Wentworth scale[26].

**Reporting summary**
Further information on research design is available in the Nature Portfolio Reporting Summary linked to this article.

## Data availability
All the data are provided in the paper's main text and Supplementary Figs. 1–7 and Supplementary Data 1–8.

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

## Acknowledgements

The authors thank UiT, The Arctic University of Norway, the Norwegian Research Council through the projects AKMA, Advancing Knowledge of Methane in the Arctic (project number 287869), HOTMUD (project number 288299), NCS2030 (project number 331644), WELLFATE (project number 344447), and EMAN7 (project number 320100), the Norwegian Offshore Directorate, REV Ocean, Woods Hole Oceanographic Institution, and La Rochelle University for their financial support and facilitation of the research expeditions. TGS is thanked for permission to use the 3D-seismic data SPE16M01 shown in Supplementary Fig. 1. We are grateful to Nicolas Straube and Ingvar Bykjedal at the Department of Natural History, University Museum of Bergen, for their help with the identification of fishes, to Alexandra Padilla at Woods Hole Oceanographic Institution for supporting the SAGE data analyses, to Tone Tjelta Hansen at the Norwegian Offshore Directorate for the preparation of the microfossils, to Lawerence Hislop at REV for the support with ROV images, Heike Jane Zimmerman for the illustrated infographic and Pär Gunnar Jansson for his guiding light during the AKMA 3 cruise. The REV Ocean dive team's invaluable contributions and expertise in data collection are acknowledged with gratitude. SPR work was supported by FCT/MCTES in the scope of the CEEC contract (CEECIND/00758/2017) and funds attributed to CESAM (UIDP/50017/2020, UIDB/50017/2020 and LA/P/0094/2020). MHE was funded by the Trond Mohn Foundation through the Centre for Deep Sea Research (grant number TMS2020TMT13) and the Norwegian Biodiversity Information Centre (the Taxonomy Initiative) through the project "Fauna of hydrothermal vents and cold seeps in Norwegian waters" (project number 3-20-70184243). IBA is funded by the Swiss National Science Foundation (project number: 221959). The Nippon Foundation is acknowledged for supporting the Ocean Census programme, whose scientists participated in this expedition and the subsequent identification of fauna.

## Author contributions

G.P. designed the study. G.P., C.A., A.S, B.F., M.H.E., A.D.R., F.H., S.B., A.P.M., R.M. S.P.R., A.M. analysed the data and wrote the manuscript. A.S. and F.H. performed video data analyses. C.A. performed geochemical interpretations. B.F. performed water column analyses. M.H.E., A.D.R., S.P.R. and D.S. carried out megafauna analyses. T.E. performed the stratigraphic interpretations. S.Y., B.C.C., A.P.M.M., J.A.K. analysed SAGE methane data. I.B.A performed eDNA analyses. A.M., S.P. and R.M. supported the geological interpretations. D.K. and S.B. contributed to data acquisition. All co-authors commented on the manuscript.

## Funding

## Competing interests

The authors declare no competing interests.
