## [Peer Review File · Nature Communications]

Sanctuary for vulnerable Arctic species at the Borealis Mud Volcano

Corresponding Author: Professor Giuliana Panieri

Version 0:

Reviewer comments:

Reviewer #1

(Remarks to the Author)

The manuscript entitled "The discovery of Borealis mud volcano: a natural sanctuary for threatened Arctic species" discusses the main characteristics of Borealis, a submerged mud volcano system located in Norwegian waters. It conducts a comprehensive characterization that covers geological, geochemical, and biological components. The study makes a novel and significant contribution, though it seems to primarily interest a specific and localized research community. It is important to note that mud volcanoes are prevalent across various global oceanic regions. A quick search on Web of Science using the terms 'mud volcano' and 'sea' yielded approximately 700 articles, underscoring the ecological significance of these structures. However, the discussions in the study focus mainly on other cold seeps in the Arctic region, leaving the potential influence or benefits of the results on the international scientific community uncertain.

The manuscript requires revision. The title promotes the mud volcano as a 'natural sanctuary for threatened Arctic species', yet among the species observed, only one (*Sebastes norvegicus*) is listed as endangered. The authors note the presence of species indicative of Vulnerable Marine Ecosystems, such as octocorals (*P. resedaeformis*), but report that colonies of *P. resedaeformis* are 'sparse' and located only in areas with irregular carbonate formations where there is little or no sediment deposition. Therefore, the use of 'threatened' in the title appears sensationalist and should be reconsidered.

The methods should be presented after the introduction to enhance the understanding of the results and discussions.

The study does not specify the criteria for selecting sampling areas or the number of samples collected. It also lacks details on the statistical sampling procedures that would allow for an adequate characterization of the studied area. Statements, such as those in lines L129-132, are emphatic and seem unsupported by the sampling effort, as the conclusions were based on just two samples. This raises concerns about whether the sampled area is representative of the mud volcano's influence area. Additionally, seasonal aspects that affect the occurrence and distribution of species should be considered. Specifically, Figure 2 in lines L205-209 does not support the statement that 'large schools of saithe and various demersal species are clustering around the irregular carbonate formations.' The discussion concerning the fish population in the area, particularly in line L211, questions whether there is actual support for local fish populations, given that most of the identified species undergo seasonal migrations between coastal and offshore regions, suggesting that the mud volcano may simply be a transit zone rather than a vital habitat as suggested. Moreover, when discussing aspects of fish fauna and other benthic organisms, the study fails to present quantitative data on abundance or density, which is critical for a detailed understanding of these species' ecological impact in the studied area. This absence of quantitative data further compromises the robustness of the conclusions drawn about the ecological role of the mud volcano as a habitat.

Analyses of oil geochemistry fall outside the scope of my expertise, and I am therefore unable to fully assess this aspect of the study.

Other specific comments (minor comments) are listed below:

- In Figure 1a, include a map that more clearly identifies the region highlighted in the Polar North Atlantic.
- L124-125: What does "medium-sized sand" mean, and what is implied by "minor clayey content" and "few small sedimentary rock fragments"? These terms seem very subjective and not specific enough.
- L170: The referred table does not contain information associated with that result, i.e., species composition.
- L173: Was any measure of diversity estimated, or are the authors referring to the number of species? How is it relatively

low compared to what?

- L220-221: Where is this VMS information?

- L418-419: Were the images completely analyzed (i.e., 14h and 4 min of images) or were only fractions of the videos analyzed?

- L419-421: More details are needed regarding these physical samples and analyses (e.g., number of samples, volume, sampled areas, sample processing, etc).

- L424: Which supplement is referred to as "Supp.X"?

- L434: The citation format is different from the rest of the text.

- L442: Change "Fiog.1_SI" to "Fig.1_SI"

- L462-472: The citation format is different from the rest of the text.

Reviewer #2

(Remarks to the Author)

The paper presents a broad overview of a new submerged mud volcano, Borealis. It is my impression that this general paper will be followed up by many specialized papers of each of the authors. This paper focuses on the geophysical and geochemistry description of the volcano. Paleomicrobiology ages the rising sediments as Pleistocene. The organic geochemistry of the sediments and gas emissions suggest ancient organic sources that were subjected to long periods of microbial degradation. The article presents a geological history that strongly supports a deglaciation trigger model or a hot fluids surge for the origin of the volcano.

Borealis is the second mud volcano to be identified in the Barents Sea. The paper successfully conveys the importance of studying our seafloor and discovering features that can contribute to climate change or serve as vital ecosystem resources. The Borealis mud volcano does both. CTD measurements indicate that methane is reaching the sea surface, unlike many known marine methane features. This discovery causes one to wonder if there are other marine structures contributing methane to the atmosphere that have not found yet.

The high methane and sulfide concentrations appear to be limiting the macrofauna diversity at the site, however many types of anemones, serpulids, demosponges, nudibranchs, and octocoral colonies live on the carbonate slabs, creating a reef-like environment. This reef welcomes many fish species, including the endangered *Sebastes norvegicus*. The authors speculate that the environment's warm temperature and carbonate slabs shelter and promote reproductive success. The authors suggest that the ecosystem may enhance the breeding of many species, claiming, the potential nursery site as an asset to the fishing grounds of the Barent Sea and worth protecting.

Overall, the article is well-written and will be an interest to many deep-sea geobiologists. Methods are appropriate and standard to those in our field. The finding of the mud volcanic system is significant considering the volume of methane it is releasing and the habitat it create.

I have very few edits.

Abstract: I don't think it is necessary to start the abstract with "this paper presents". This may be a personal preference, but I would start with "Borealis is a recently discovered submerged mud volcano system..."

Section 5.2 suggests that there is a foraminiferal species list. It is labeled as X and no table was included.

Figure 1. It is my interpretation of the legend that the diamonds mark where the CTD sampled the water column, and the color of the diamond represents CH₄ concentrations. However, all the diamonds are the same color and don't reflect the concentrations in Table SI 3. Please correct the figure or make the explanation of the diamonds clearer.

Figure 2. I would note the colors of the comparison sites before the description of 2.a as the comparisons are made in several plots. Additionally, the Mosby Modu Volcano isn't charted on 2.a, which is confusing when it is listed under 2.a.

Table SI 3. The formatting of this table causes the header to be on the second page. It would be ideal if the entire table fit onto one page.

Version 1:

Reviewer comments:

Reviewer #2

(Remarks to the Author)

I appreciate the comments and concerns of reviewer 1 and the changes that the writers have made to address all the comments in general. I still have a few suggestions.

line 32. What is I-oxygen?

Line 33. I agree with Reviewer #1, I don't think one expedition can ensure the reasons why fauna were found at this site. Maybe word the sentence "carbonate structures may serve as"

line 60. Is there a period after organisms(4)? If so, please remove.

Figure 1. I still do not understand the relevance of the diamond colors. I appreciate that the methane concentrations are written to the side. Maybe make the diamonds all one color and depend only on the written values. Additionally, what is the significance of the gas flare colors? Are the temperatures? If so, were is the legend for that?

Reviewer #3

(Remarks to the Author)

Dear authors, thank you for a great paper of a important discovery of a mud volcano system "Borealis" located in the North Atlantic (Bear Island Trend). I have only reviewed the faunistic part of this manuscript, and understand that a following up paper will be made for the fauna.

For this manuscript I would recommend to clarify the points I identified so that the reader will feel faunistic convinced at this preliminary stage, and should look forward for a more "in depth" paper that will come and will apply the evidence (lacking in this manus) for the described observations in this paper.

The review of the manuscript entitled "The discovery of Borealis mud volcano: a natural sanctuary for vulnerable Arctic species" (NCOMMS-24-20285A).

Giuliana Panieri, Claudio Argentino, Alessandra Savini, Bénédicte Ferré, Rune Mattingsdal, Sofia Ramalho, Mari Eilertsen, Tor Eidvin, Sarah Youngs, Beckett Colson, Anna Michel, Jason Kapit, Fereshteh Hemmateenejad, Denise Swanborn, Alex Roger, Adriano Mazzini, Ines Barrenechea Angeles, Stephane Polteau, Dimitri Kalenitchenko, and Stefan Buenz

I will only review the "biological" aspects of this work.

This manuscript describes a mud volcano system "Borealis" located in the North Atlantic (Bear Island Trend). Methane-dominated fluids are eruption of from a localized site within a 500 m diameter crater. The seafloor around comprises laterally extensive carbonate deposits.

Sampling and seafloor images reveal that Borealis hosts unique habitats thriving in the environments around methane seeps. Additionally, the irregular shaped carbonate structures serve as a substratum for sessile fauna and function as nursing grounds for threatened fish species. These carbonate structures are also a natural refuge from bottom trawling. This highlights the ecological value of cold seep ecosystems, which play a critical role in biodiversity by acting as sanctuaries for marine species, hence emphasizing the importance of their conservation.

- What are the noteworthy results?

A newly discovered mud volcano named Borealis mud volcano located in Outer Bjørnøyrenna at 390 m water depth in the Barents Sea.

Within the craters dense and extensive patches of microbial mats and tubeworm aggregations (*Oligobrachia* sp.) are observed which seem to be the main characteristic of high-latitudes compared to lower-latitudes cold seeps. A variety of microorganisms/multitude of foraminifera species were obtained from environmental DNA (eDNA) analysis (hard-shelled foraminifera (*Epistominella* sp., *Reophax dentaliniformis*, *Stainforthia* sp.), monothalamids and soft-shelled foraminifera nestled within these microbial ecosystems.

These preliminary eDNA analyses suggest the possibility of previously unidentified species potentially unique to those type of environments.

A observed scarcity of megafauna diversity suggests that the immediate environmental conditions surrounding Borealis may be inhospitable for a broader range of organisms (elevated methane concentrations, extensive carbonate crusts on the seafloor and the suspended sediment particles emitted by the gryphon).

ROV imagery revealed that the megafauna is dominated by dense clusters of anemones and serpulids anchored to carbonate substrates. *Tubularia* sp. colonies flourish on the crater slopes appearing resilient to the presence of suspended sediment. Cladorhizid sponges and sea stars is more sporadic.

Sample collections (possible taken by a push core, but not explained) from various habitats within the Borealis yielded taxa not detectable via ROV imagery, such as annelids, amphipods, gastropods, polyplacophorans, nemerteans (*Nipponemertes* spp.), and ophiuroids. The results from these samples are not included in this manuscript but will be elucidated later.

The extensive carbonates provide additional habitat and suitable hard substratum for erect epifauna and dense aggregations of several species of anemones, serpulids, demosponges, nudibranchs, and sparse octocoral colonies.

It was annotated by the ROV videos that the corals *P. resedaeformis* was only located in the jagged carbonate area with lower or no sediment deposition from the plumes of the gryphon was evident.

Large schools of saithe (*Pollachius virens*) and various demersal species such as spotted wolffish (*Anarhichas minor*), cod (*Gadus morhua*), four bearded rockling (*Enchelyopus cimbrius*), and redfish (mainly *Sebastes norvegicus* but a few individuals tentatively identified as *S. viviparus* and *S. mentella*) clustering around the jagged carbonate formations suggest that these structures serve as critical habitats, offering both shelter and abundant feeding opportunities, thereby playing a pivotal role in sustaining the local fish populations.

Several individuals of the red-listed *S. norvegicus* observed in the ROV videos appear to be pregnant.

The morphology of the carbonates providing a protective shelter, and the site's elevated temperatures are said to possible enhance reproductive success by accelerating egg development.

Lost fishing gear on the carbonate rock around the perimeter of the craters was observed by ROV imagery. The gear was heavily colonised by sessile fauna (e.g., anemones and hydrozoans), suggesting being old. No recent fishing activities (VMS and AIS, OSPAR/ICES) are recorded around the mud volcano system.

The presence of red-listed *S. norvegicus* and VME octocorals suggests that the Borealis is a sanctuary for endangered species.

The absence of bottom trawling and the natural habitat's protective qualities offer a refuge where these species can thrive despite significant seafloor impacts from fishing in the surrounding Barents Sea.

- Will the work be of significance to the field and related fields? How does it compare to the established literature? If the work is not original, please provide relevant references.

Despite numerous observations of methane emissions from the seafloor in Arctic regions, only five mud volcanoes have been discovered in the Canadian Beaufort Sea (Western Arctic), and three in Alaska, and so far, the Håkon Mosby Mud Volcano was the only known structure in Norwegian waters.

This work is significant to the field because it is an extremely important new and seldom discovery of a mud volcano that will need to be registered on sea-maps as soon as possible to avoid damaging this complex habitat and its associated fauna. Comparison to established literature is good, but there are issues that need to be taken care of.

- Does the work support the conclusions and claims, or is additional evidence needed?

Lines 215-218

Saithe (*Pollachius virens*), spotted wolffish (*Anarhichas minor*), cod (*Gadus morhua*), bearded rockling (*Enchelyopus cimbrius*) can be identified on images. But splitting redfish into *Sebastes norvegicus*, *S. viviparus* and *S. mentella* is challenging and would need more verification to become trustworthy (e.g., to be taken by physically by equipment and identified by experts). This would need to be explained in the text.

Lines 223-225: Several individuals of the red-listed *S. norvegicus* observed in the ROV videos appear to be pregnant. This is written without any references or further description. As above, this would need much more evidence since it is extremely difficult (impossible) to judge if a *S. norvegicus* is pregnant from a picture.

Lines (226-228) 287-290: The elevated temperatures around the seepage sites may create optimal conditions for reproductive processes, leading to more effective breeding cycles. This thermal advantage can enhance the survival and growth of fish populations in the deep sea, and the Barents Sea, one of the world's most productive fishing grounds, represents a significant ecological asset.

It is highly unlikely that a 500m diameter spot can have an effect on the "fish-population" in the Barents Sea. It may have an impact on the local fish using this area for spawning. This is written without any references or further evidence and need to be more carefully evaluated.

Lines 216-221: Large schools of fish are clustering around the jagged carbonate formations suggest that these structures serve as critical habitats, offering both shelter and abundant feeding opportunities, thereby playing a pivotal role in sustaining the local fish populations.

This is built on evidence from observations in *Lophelia* reefs and artificial reef-life structures – but I am missing evidence that the pelagic prey that the mention fish species are eating are found at the volcano structures. Did you register these pelagic prey-species on your video-records?

Lines 169-204

It is very hard to understand what was observed via ROV imagery, what was sampled using a push core, and how sediment was taken for DnA analyses. I question is it may be too premature to include the faunistic information in this paper since it anticipated that another and likely more describing paper will be made by the authors.

Lines 187-192: Anemones, annelids, amphipods, gastropods, polyplacophorans, and ophiuroids.

These large animal groups are found everywhere and does not really provide any information. I may suggest describing their traits or morphological characteristic in this paper instant, and then in next paper provide more detailed taxonomy.

- Are there any flaws in the data analysis, interpretation and conclusions? - Do these prohibit publication or require revision? Some sentences need to be strengthened and definitions clarified before publication:

Lines 352-374

A photogrammetry technique (SfM) was applied and selected ROV video frames provided physical habitats and sedimentary facies. But are these pictures also analyzed for fauna – please explain?

The substrate was differentiated in carbonate crusts (CC), Microbial Mats (MM) and all other substrate (denoted as BG in figure 4, but not as such in the text, and this is confusing).

Macrofauna occurrences were classified as Red Fishes (RedF) and Coral (CC). The Coral category are denoted "CC" in line 366 (but not in figure 4), -the same as carbonate crusts – and this is confusing.

Each ROV dive surveyed different areas, with some level of overlapping between ROV tracklines crossing the Borealis MV main seeping sites. – Does this represent a danger of counting and recording of observations more than ones?

The surveyed area covered was estimated by using GIS-based tool and applying a 1 m² buffer area surrounding each ROV position. Because of the differences in the extent of the areas surveyed by each video, the abundance of the detected macrofauna (Red Fishes and Coral) are expressed as number per 10 square meter.

- Is the methodology sound? Does the work meet the expected standards in your field?

The analyzing of fauna from video (and maybe images?) is an accepted and well used method. The abundance of observed fauna is standardized.

But the paper would be clearer and more convincing if explained and verified more deeply (see above). I guess that this will be the case for the next paper wish I will look very much forward to.

- Is there enough detail provided in the methods for the work to be reproduced?

There are not enough details provided for the fauna-work to be reproduced. I guess that this will be the case for the next paper wish I will look very much forward to.

REBUTTAL LETTER

We thank the reviewers for their thorough reading of our manuscript and their constructive comments. Below we have copied each review in full (in black text), and highlighted (main) reviewer comments in **black bold** text. We provide our response to them in Blue text.

Thanks to these requested comments and suggestions, we feel the manuscript has improved considerably and hope that our proposed revision will meet the criteria for publication in ***Nature communications***.

Sincerely,
Giuliana Panieri (on behalf of all authors)

REVIEWER COMMENTS

Reviewer #1 (Remarks to the Author):

Reviewer: The manuscript entitled “The discovery of Borealis mud volcano: a natural sanctuary for threatened Arctic species” discusses the main characteristics of Borealis, a submerged mud volcano system located in Norwegian waters. It conducts a comprehensive characterization that covers geological, geochemical, and biological components. The study makes a novel and significant contribution, though it seems to primarily interest a specific and localized research community. It is important to note that mud volcanoes are prevalent across various global oceanic regions. A quick search on Web of Science using the terms 'mud volcano' and 'sea' yielded approximately 700 articles, underscoring the ecological significance of these structures. **However, the discussions in the study focus mainly on other cold seeps in the Arctic region, leaving the potential influence or benefits of the results on the international scientific community uncertain.**

Response: We acknowledge the reviewer for the comments and the opportunity to clarify the broader implications of our study on the Borealis mud volcano. We appreciate the recognition of the novelty and significance of our research within its specific field, and we have now improved our manuscript quite substantially. In response to the reviewer’s concerns regarding the focus of our study and its appeal to the international scientific community, we would like to clarify the followings:

- while our manuscript does concentrate on the Borealis mud volcano in the Arctic, our findings have broader implications that extend beyond this area in the Barents Sea. Mud volcanoes, as the reviewer rightly pointed out, are widespread across various regions. However, there are not many papers dealing with biological characteristics of these features. We reported remarkable observations of the biological characteristics from this site and for this resubmission, we have also added the abundance of the vulnerable main macrofauna species strengthening the hypothesis that those sites represent oasis for vulnerable species.
- The geological, geochemical, and biological characteristics observed at Borealis can provide a comparative dataset that increase the understanding of similar structures worldwide. It is important to acknowledge, as we briefly mentioned earlier, the scarcity of comprehensive data on many mud volcanoes and their biology. As pointed out by the reviewer, it is true that a search on Web of Science using the terms 'mud volcano' and 'sea' yielded approximately 700 articles, but a limited number of them (ca. 20) report on their ecological significance, which is what we aim with our work. It needs to be mentioned that the scarcity of comprehensive data (from geophysics, geochemistry and biology), is largely due to the logistical challenges and significant resources required to conduct such studies. These environments are difficult to access and require research vessels and specialized equipment. This reality underscores the value and necessity of our detailed study at Borealis, which contributes data to a field where comprehensive information is still limited and might trigger international collaborations for the investigations of similar sites.

- The ecological role of Borealis in supporting threatened species is a critical aspect of our study that has an environmental relevance that go beyond the site itself. As the Reviewer 2 pointed out “This discovery causes one to wonder if there are other marine structures contributing methane to the atmosphere that have not found yet”. The comment is suggesting that similar marine structures could play unrecognized roles in both biodiversity support and climate dynamics. We agree with the reviewer, and we are positive that the discovery of Borealis might trigger other similar discovery in the Arctic in the future now that we are aware of this geological feature.

We think that our arguments and the additions to the text will make the significance of our findings clearer to a broader audience and underscore the global importance of studying such unique ecological systems.

Reviewer: The manuscript requires revision. **The title promotes the mud volcano as a 'natural sanctuary for threatened Arctic species', yet among the species observed, only one (Sebastes norvegicus) is listed as endangered.** The authors note the presence of species indicative of Vulnerable Marine Ecosystems, such as octocorals (*P. resedaeformis*), but report that colonies of *P. resedaeformis* are 'sparse' and located only in areas with irregular carbonate formations where there is little or no sediment deposition. **Therefore, the use of 'threatened' in the title appears sensationalist and should be reconsidered.**

Response: We acknowledge the reviewer for the comments. We understand the reviewer's concerns about the use of the term "threatened" in the title and the representation of the species observed at the Borealis mud volcano. The term "threatened" might convey an unintended sense of urgency or sensationalism, as the reviewer has pointed out. To address this concern and to more accurately reflect the content of our study, we propose revising the title to: "The Discovery of Borealis Mud Volcano: A Refuge for Vulnerable Arctic Marine Species". We think that the revised title maintains the integrity and focus of our research while addressing the concerns about the representation of the ecological status of the species observed. It also aligns with the data presented in the manuscript, emphasizing the role of Borealis as a habitat for species within Vulnerable Marine Ecosystems without implying a broader threat status than is supported by our findings.

Reviewer: The methods should be presented after the introduction to enhance the understanding of the results and discussions.

Response: We followed the template provided by Nature communications but we are willing to change the order of the chapters if the editor ask to do to.

Reviewer: The study does not specify the criteria for selecting sampling areas or the number of samples collected. It also lacks details on the statistical sampling procedures that would allow for an adequate characterization of the studied area. **Statements, such as those in lines L129-132, are emphatic and seem unsupported by the sampling effort, as the conclusions were based on just two samples.** This raises concerns about whether the sampled area is representative of the mud volcano's influence area. **Additionally, seasonal aspects that affect the occurrence and distribution of species should be considered. Specifically, Figure 2 in lines L205-209 does not support the statement that 'large schools of saithe and various demersal species are clustering around the irregular carbonate formations.'** The discussion concerning the fish population in the area, particularly in line L211, questions whether there is actual support for local fish populations, given that most of the identified species undergo seasonal migrations between coastal and offshore regions, suggesting that the mud volcano may simply be a transit zone rather than a vital habitat as suggested. Moreover, when discussing aspects of fish fauna and other benthic organisms, **the study fails to present quantitative data on abundance or density, which is critical for a detailed understanding of these species' ecological impact in the studied area.** This

absence of quantitative data further compromises the robustness of the conclusions drawn about the ecological role of the mud volcano as a habitat.

Response: The association of saithe with deep-water reef habitat is well established off the Norwegian coast, albeit observations have mainly concerned occurrence of this species with *Lophelia pertusa* reefs (e.g. Costello et al (2005) in Freiwald A, Roberts JM (eds) *Cold-water Corals and Ecosystems*. Springer-Verlag Berlin Heidelberg, pp 771-805; Mortensen et al (1995) *Sarsia* 80: 145-158; see evidence for also shoaling around artificial reefs [rigs] Soldal et al (2002) *ICES Journal of Marine Science*, 59: S281–S287). It is therefore unsurprising that this species is associated with irregular seafloor structures associated with broken carbonate pavement around the Borealis mud volcano. The referee is correct in that this species undergoes migrations between more inshore and offshore areas during spring (e.g. Olsen et al., (2009) *ICES Journal of Marine Science*, 67: 87–101), however, the use of such reef-like habitats during such migrations does confer shelter, perhaps foraging opportunities and protection from fishing for these species during their migratory movements. Therefore, irrespective of whether use of the habitat is temporary or longer term, protection during part of the life history is provided.

Reviewer: lines L129-132: The reviewer refers to those lines in the main text:

*“Nearly all the individuals are extinct species typically associated with Pleistocene deposits on the Norwegian Shelf from ca 700-1000 m bsf28,29. The lack of benthic foraminifera typical for Holocene and recent sediments on the Norwegian continental shelf, including *Trifarina angulosa* and *Uvigerina peregrina*30-32, and the almost complete lack of warm water-dwelling planktonic foraminifera indicate that no Holocene sediments are present in the rising sediment.”*

Response: In the specified lines, our discussion focuses on the findings related to foraminifera and biostratigraphic analyses. It is important to note that the nature of biostratigraphic research, particularly when involving foraminifera, does not typically require extensive sampling to yield significant insights and often one sample is enough to establish the age of a sediment. Foraminiferal analyses can provide valuable information on the environmental conditions and geological age, even from a limited number of samples. This is due to the fact that foraminifera are highly sensitive to their environments, and their presence, absence, or community structure can be highly indicative of specific ecological conditions or geological interval.

Furthermore, to address the reviewer comment regarding the representativeness of our sample, we would like to clarify that the analyzed samples were directly collected from the mud expelled by the volcano so it is highly indicative of the expulsion activity of material from below and provides essential data needed to characterize Borealis as a mud volcano. To illustrate this, we have included a frame from the ROV video in Figure 4_SI, which clearly shows the precise location where the samples were taken. We hope this explanation clarifies the methodology and rationale behind our sampling strategy and the conclusions drawn from our biostratigraphic analysis.

Reviewer: “seasonal aspects that affect the occurrence and distribution of species should be considered”

Response: Regarding the comment of the reviewer about seasonality, we must clarify that our current dataset is limited to observations made during the single expedition that discovered Borealis MV. As such, we do not have other observational data that allow us to evaluate seasonal variations in species behaviour or distribution around Borealis. We must admit that we did not think about this earlier, but definitely Borealis presents an ideal location for establishing a permanent monitoring station to continuously measure methane emissions and other environmental parameters, in addition to observe fish migrations. In any case, we answered to this point in the text above while commenting on the general comment of the reviewer.

Reviewer: Moreover, when discussing aspects of fish fauna and other benthic organisms, **the study fails to present quantitative data on abundance or density, which is critical for a detailed understanding of these species' ecological impact in the studied area.** This absence of quantitative data further compromises the robustness of the conclusions drawn about the ecological role of the mud volcano as a habitat.

Response: In response to the reviewer your concerns, we would like to clarify that the primary focus of our initial study was to establish a baseline presence and distribution of species associated with the mud volcano Borealis habitat, providing a data set (from geophysics through biostratigraphy and geochemistry) that clarify the nature of this unique geological feature for the Polar North Atlantic. Given the exploratory nature of this research, our initial approach regarding all the macrofauna from this site was qualitative. However, we recognize the critical importance of quantitative assessments in ecological studies, as the reviewer pointed out, and agree that such data are essential for a more robust analysis of the ecological roles of these species. To address this gap, we are planning more than one follow-up studies that will specifically target quantitative measurements of species abundance and density. However, because we acknowledge the concern of the reviewer, we viewed all the ROV dives (ROV 13, ROV 14, ROV 15 and ROV 16) and annotated the presence of all the main vulnerable macrofauna occurrences and produced a matrix as described in the Method section in the chapter 3. ROV and video analyses and included in Table 8_SI (Table 8_SI: Abundance of the vulnerable main macrofauna species observed at Borealis MV). We then produced histograms showing the abundance of main vulnerable macrofauna occurrences (RedF: Red Fishes; CC: Coral), identified through video analyses and expressed as the number of occurrences/10 m², per substrate (BG: Sedimented background; MM: Microbial Mat; CC: Carbonate Crust) and each video (ROV 13: ROV 14; ROV 15; ROV 16) and an histogram showing the abundance of the main vulnerable macrofauna occurrences (Red Fishes and Coral), identified through video analyses and expressed as the number of occurrences/10 m², considering the total visualized area per each substrate (MM: Microbial Mat; CC: Carbonate Crust; BG: Background). Now those histograms are included in the main text as Fig. 4. As we have now explained in the text, we observed that red fish are consistently more abundant around the carbonate crusts, exhibiting occurrence rates ranging from approximately 0.02 to 0.76 per 10 square meters. These numbers, when compared with the background fish populations investigated around Borealis from sedimented background where the number are basically 0, clearly indicate that the fish preferentially inhabit areas around the carbonate crusts. Additionally, corals (which are also vulnerable species, as described in the main text) are exclusively found on the carbonate crusts, with occurrence rates varying from 0.09 to 0.03 per 10 square meters. Those type of corals do not live on sediment seafloor.

Reviewer: Analyses of oil geochemistry fall outside the scope of my expertise, and I am therefore unable to fully assess this aspect of the study.

Response: We cannot comment this point, but we are confident of our work.

Other specific comments (minor comments) are listed below:

Reviewer:- In Figure 1a, include a map that more clearly identifies the region highlighted in the Polar North Atlantic.

Response: We acknowledge the comment of the reviewer, and the Figure 1 is now clearer and identify better the area of investigation in the Polar North Atlantic.

Reviewer: - L124-125: What does “medium-sized sand” mean, and what is implied by “minor clayey content” and “few small sedimentary rock fragments”? These terms seem very subjective and not specific enough.

Response: To address your concerns about the terms "medium-sized sand," "minor clayey content," and "few small sedimentary rock fragments," we provide clarifications that have been included in the

section “Methods” in the subchapter 6. Biostratigraphy and sedimentology where we indicated the classification used (Wentworth scale, also included in the reference list) for the definition of the term used to describe the sediment emitted from Borealis. In addition, we also added information of the Biostratigraphy, that was non originally included.

Reviewer: - L170: The referred table does not contain information associated with that result, i.e., species composition.

Response: We appreciate the reviewer's attention in identifying this error. We now added the Table 7_SI with the foraminifera list identified from eDNA analyses with the following detail: Table 7_SI: Amplicon Sequencing Variants (ASV) of foraminiferal eDNA. For each ASV the foraminifera taxon identified on a number of reads > 100 are indicated.

Reviewer: - L173: Was any measure of diversity estimated, or are the authors referring to the number of species? How is it relatively low compared to what?

Response: In this respect, in the text we refer to the observed number of species based on visual inspections from ROV video footage, rather than a calculated measure of diversity such as Shannon or Simpson indices that are planned. The term "relatively low" was used to qualitatively describe the observed species richness in comparison to the background area and other similar habitats. To clarify this in our manuscript, we revised the text to specify that our reference to diversity is based solely on observed species trend and that these observations are not backed by statistical analyses (that are planned for future work).

Reviewer: - L220-221: Where is this VMS information?

Response: We now included in the text the reference to the Vessel Monitoring System (VMS) data that are reported in a well-known publication from the MOREANO project and the EU-based bottom fishing intensity data from the OSPAR/ICES database, that we used to extract GIS data to confirm that no recent fishery has occurred in the area where Borealis is located. We also confirmed with Automatic Identification System (AIS) data from the Global Fishing Watch database, to double check from the 2020s onwards data, and the same output, no fishing there: <https://globalfishingwatch.org/map/index?latitude=72.99823215175326&longitude=12.510194070543434&zoom=3.7222661775128367&start=2024-03-09T00%3A00%3A00.000Z&end=2024-06-09T00%3A00%3A00.000Z>. All these information have been added in the text.

Reviewer: - L418-419: Were the images completely analyzed (i.e., 14h and 4 min of images) or were only fractions of the videos analyzed?

Response: We thank the reviewer for the comment, that gave us the possibility to clarify the work we did on all the videos. In Table 1_SI (reporting the type of sample and analyses/measurement performed on each) we indicated the entire duration of each ROV video (in the column “note”) and all the videos were completely analysed not fractions of them.

Reviewer: - L419-421: More details are needed regarding these physical samples and analyses (e.g., number of samples, volume, sampled areas, sample processing, etc).

Response: We thank the reviewer for the comment that allow us to clarify those aspect of our work, as we have also indicated in the main text in the sub chapter 5.1 Mega/macrofauna under 5. Biological samples and observations. During this initial phase of our research, not all collected samples were analyzed. Our primary goal with this paper was on those samples that would most likely provide immediate clarification and support for the video-based identifications. A more detailed and comprehensive investigation of all collected samples is planned for October as part of our ongoing research in a Workshop funded by Ocean Census. This future work will include a thorough analysis of

the remaining samples, which will give us the possibility to write one (or more) paper with the details suggested by the reviewer

Reviewer: - L424: Which supplement is referred to as “Supp.X”?

Response: We thank the reviewer for the comment. We now added the Table 7_SI with the foraminifera list identified from eDNA analyses with the following detail: Table 7_SI: Amplicon Sequencing Variants (ASV) of foraminiferal eDNA. For each ASV the foraminifera taxon identified on a number of reads > 100 are indicated.

Reviewer: - L434: The citation format is different from the rest of the text.

Response: We thank the reviewer for the comment. Now the citation is formatted following the Nature comm guidelines.

Reviewer: - L442: Change “Fiog.1_SI” to “Fig.1_SI”

Response: Corrected as suggested.

Reviewer: - L462-472: The citation format is different from the rest of the text.

Response: We thank the reviewer for the comment. Now all the citations that were included in the caption of Fig. 3 are formatted following the Nature comm guidelines.

Reviewer #2 (Remarks to the Author):

The paper presents a broad overview of a new submerged mud volcano, Borealis. It is my impression that this general paper will be followed up by many specialized papers of each of the authors. This paper focuses on the geophysical and geochemistry description of the volcano. Paleomicrobiology ages the rising sediments as Pleistocene. The organic geochemistry of the sediments and gas emissions suggest ancient organic sources that were subjected to long periods of microbial degradation. The article presents a geological history that strongly supports a deglaciation trigger model or a hot fluids surge for the origin of the volcano.

Borealis is the second mud volcano to be identified in the Barents Sea. The paper successfully conveys the importance of studying our seafloor and discovering features that can contribute to climate change or serve as vital ecosystem resources. The Borealis mud volcano does both. CTD measurements indicate that methane is reaching the sea surface, unlike many known marine methane features. This discovery causes one to wonder if there are other marine structures contributing methane to the atmosphere that have not found yet.

The high methane and sulfide concentrations appear to be limiting the macrofauna diversity at the site, however many types of anemones, serpulids, demosponges, nudibranchs, and octocoral colonies live on the carbonate slabs, creating a reef-like environment. This reef welcomes many fish species, including the endangered *Sebastes norvegicus*. The authors speculate that the environment’s warm temperature and carbonate slabs shelter and promote reproductive success. The authors suggest that the ecosystem may enhance the breeding of many species, claiming, the potential nursery site as an asset to the fishing grounds of the Barent Sea and worth protecting.

Overall, the article is well-written and will be an interest to many deep-sea geobiologists. Methods are appropriate and standard to those in our field. The finding of the mud volcanic system is significant considering the volume of methane it is releasing and the habitat it create.

I have very few edits.

Reviewer: Abstract: I don't think it is necessary to start the abstract with "this paper presents". This may be a personal preference, but I would start with "Borealis is a recently discovered submerged mud volcano system..."

Response: We have followed the suggestions of the reviewer and changed the text as suggested. Now the abstract start with "Borealis is a recently discovered submerged mud volcano system....."

Reviewer: Section 5.2 suggests that there is a foraminiferal species list. It is labeled as X and no table was included.

Response: We thank the reviewer for the comment. We now added a Supplementary Table 7 with the foraminifera list identified from eDNA analyses and the new table name is Supplementary Table 7: Amplicon Sequencing Variants (ASV) of foraminiferal eDNA. For each ASV the foraminifera taxon identified on a number of reads > 100 are indicated.

Reviewer: It is my interpretation of the legend that the diamonds mark where the CTD sampled the water column, and the color of the diamond represents CH4 concentrations. However, all the diamonds are the same color and don't reflect the concentrations in Table SI 3. Please correct the figure or make the explanation of the diamonds clearer.

Response: We thank the reviewer for the comment. We modified Figure 1 according to the suggestions and made it better and more readable than before. We add the CTD and SAGE Methane concentration data in a zoomed circle where the concentration can be easily seen and added also the number to better appreciate the results of the measurements.

Reviewer: Figure 2. I would note the colors of the comparison sites before the description of 2.a as the comparisons are made in several plots. Additionally, the Mosby Modu Volcano isn't charted on 2.a, which is confusing when it is listed under 2.a.

Response: We think that the suggestions of the reviewer refers to Figure 3. As the reviewer suggested, we moved the comparison to the colours characterizing the different sites in several plot in the first part of the Figure description and adjusted the captions.

Reviewer: Table SI 3. The formatting of this table causes the header to be on the second page. It would be ideal if the entire table fit onto one page.

Response: All the tables have been reformatted according to "Nature communication guide to formatting articles". Now there are 4 Tables (Supplementary Table 3-6) instead of Table 3_SI reporting the SAGE data.

We thank the reviewers for their thorough reading of our manuscript and their constructive comments. Below we have copied each review in full (in black text) and highlighted (main) reviewer comments in **black bold** text. We provide our response to them in blue text.

Thanks to these requested comments and suggestions, we feel the manuscript has improved considerably from the previous version submitted and hope that our proposed revision will meet the criteria for publication in ***Nature Communications***

Sincerely,

Giuliana Panieri (on behalf of all authors)

REVIEWER COMMENTS

Reviewer #2 (Remarks to the Author):

I appreciate the comments and concerns of reviewer 1 and the changes that the writers have made to address all the comments in general. I still have a few suggestions.

line 32. What is I-oxygen?

We corrected the text and changed it to low oxygen.

Line 33. I agree with Reviewer #1, I don't think one expedition can ensure the reasons why fauna were found at this site. Maybe word the sentence "carbonate structures may serve as"

The concern raised by Reviewer #1 has been better explained in the revised version of the text in the chapter "A unique oasis for faunal communities" also following the suggestions of Reviewer #3. Regarding this comment from Reviewer #2, we changed the sentences suggested and added the word "may" to the sentence.

line 60. Is there a period after organisms(4)? If so, please remove.

The period has now been removed as suggested.

Figure 1. I still do not understand the relevance of the diamond colors. I appreciate

that the methane concentrations are written to the side. Maybe make the diamonds all one color and depend only on the written values. Additionally, what is the significance of the gas flare colors? Are the temperatures? If so, were is the legend for that?

We appreciate the reviewer's comment, which prompted us to enhance Figure 1 and its caption. The diamond symbol in the figure represents the CTD samples measured for methane concentration. Since these samples reflect methane concentration measurements in water, similar to the SAGE data, we have used the same color scale for both, to maintain consistency. Additionally, we have included a zoomed-in view of the depression where methane measurements were conducted to improve clarity. To facilitate the reader's appreciation of Figure 1, we labelled all figures using the a, b, c convention and provided a more detailed explanation in the caption.

Regarding the significance of the gas flare colors. The flares represent the backscattering intensity of the reflected acoustic signals (red indicating the highest values and light green the lowest). However, the colour variations only help assess the distribution of the gas bubbles within the water column but is not indicative of methane concentration. This is a standard way of visualising acoustic data to facilitate initial analysis and guide further detailed investigations.

The revised version of the caption is the following:

Figure 1. Borealis MV. **a** Map showing the location of Borealis mud volcano and other cold seeps in the area (for some of them, the origin of emitting methane is shown in Fig. 3). **b** Active gryphon emitting warm fluid, methane and Neogene sediments. **c** Compiled observations, including seabed topography from high-resolution multibeam data (5 m grid cell), a seismic cross-section from 3D-seismic dataset SPE16M01 (the complete seismic section is available in Supplementary Figure 1) and the multi-beam echosounder data (320 kHz) tracing streams of gas bubbles (gas flares) in the water column with variations in the colours of flares representing the backscattering intensity of the reflected acoustic signals (red indicating the highest values and light green the lowest). **d** Georeferenced detail of the confined depression (~0.14 km²) around the active gryphon showing the methane water concentration (represented with the same colour scale to maintain consistency) measured in CTD water samples (position of the water sampling indicated by the diamonds in the vertical line) and real-time SAGE measurements (data showed as ROV tracks) (Supplementary Tables 3-6). **e** Georeferenced ROV images show a carbonate pinnacle colonised by *Octocorallia* (pinnacle high ca 120 cm) and **f** the red fish *Sebastes norvegicus* (ca 30 cm in length).

Reviewer #3 (Remarks to the Author):

Dear authors, thank you for a great paper of a important discovery of a mud volcano system “Borealis” located in the North Atlantic (Bear Island Trend). I have only revied te faunistic part of this manuscript, and understand that a following up paper will be made for the fauna.

For this manuscript I would recommend to clarify the points I identified so that the reader will feel faunistic convinced at this preliminary stage, and should look forward for a more “in depth” paper that will come and will apply the evidence (lacking in this manus) for the described observations in this paper.

The review of the manuscript entitled "The discovery of Borealis mud volcano: a natural sanctuary for vulnerable Arctic species" (NCOMMS-24-20285A).

Giuliana Panieri, Claudio Argentino, Alessandra Savini, Bénédicte Ferré, Rune Mattingsdal, Sofia Ramalho, Mari Eilertsen, Tor Eidvin, Sarah Youngs, Beckett Colson, Anna Michel, Jason Kapit, Fereshteh Hemmateenejad, Denise Swanborn, Alex Roger, Adriano Mazzini, Ines Barrenechea Angeles, Stephane Polteau, Dimitri Kalenitchenko, and Stefan Buenz

I will only review the “biological” aspects of this work.

This manuscript describes a mud volcano system “Borealis” located in the North Atlantic (Bear Island Trend). Methane-dominated fluids are eruption of from a localized site within a ~500 m diameter crater. The seafloor around comprises laterally extensive carbonate deposits.

Sampling and seafloor images reveal that Borealis hosts unique habitats thriving in the environments around methane seeps. Additionally, the irregular shaped carbonate structures serve as a substratum for sessile fauna and function as nursing grounds for threatened fish species. These carbonate structures are also a natural refuge from bottom trawling.

This highlights the ecological value of cold seep ecosystems, which play a critical role in biodiversity by acting as sanctuaries for marine species, hence emphasizing the importance of their conservation.

- What are the noteworthy results?

A newly discovered mud volcano named Borealis mud volcano located in Outer Bjørnøyrenna at 390 m water depth in the Barents Sea.

Within the craters dense and extensive patches of microbial mats and tubeworm aggregations (*Oligobrachia* sp.) are observed which seem to be the main characteristic of

high-latitudes compared to lower-latitudes cold seeps. A variety of microorganisms/multitude of foraminifera species were obtained from environmental DNA (eDNA) analysis (hard-shelled foraminifera (*Epistominella* sp., *Reophax dentaliniformis*, *Stainforthia* sp.), monothalamids and soft-shelled foraminifera nestled within these microbial ecosystems.

These preliminary eDNA analyses suggest the possibility of previously unidentified species potentially unique to those type of environments.

A observed scarcity of megafauna diversity suggests that the immediate environmental conditions surrounding Borealis may be inhospitable for a broader range of organisms (elevated methane concentrations, extensive carbonate crusts on the seafloor and the suspended sediment particles emitted by the gryphon).

ROV imagery revealed that the megafauna is dominated by dense clusters of anemones and serpulids anchored to carbonate substrates. Tubularia sp. colonies flourish on the crater slopes appearing resilient to the presence of suspended sediment. Cladorhizid sponges and sea stars is more sporadic.

Sample collections (possible taken by a push core, but not explained) from various habitats within the Borealis yielded taxa not detectable via ROV imagery, such as annelids, amphipods, gastropods, polyplacophorans, nemertean (*Nipponemertes* spp.), and ophiuroids. The results from these samples are not included in this manuscript but will be elucidated later.

We appreciate the reviewer's comment. In response, we have enhanced the manuscript by including detailed information on the sample collection processes in Table 1 and Section 5.1 on Mega/Macrofauna in the Methods part. This addition clarifies our work and provides transparency regarding our data collection.

The extensive carbonates provide additional habitat and suitable hard substratum for erect epifauna and dense aggregations of several species of anemones, serpulids, demosponges, nudibranchs, and sparse octocoral colonies.

It was annotated by the ROV videos that the corals *P. resedaeformis* was only located in the jagged carbonate area with lower or no sediment deposition from the plumes of the gryphon was evident.

Large schools of saithe (*Pollachius virens*) and various demersal species such as spotted wolffish (*Anarhichas minor*), cod (*Gadus morhua*), four bearded rockling (*Enchelyopus cimbrius*), and redfish (mainly *Sebastes norvegicus* but a few individuals tentatively identified as *S. viviparus* and *S. mentella*) clustering around the jagged carbonate formations suggest that these structures serve as critical habitats, offering both shelter and abundant feeding opportunities, thereby playing a pivotal role in sustaining the local fish populations.

Several individuals of the red-listed *S. norvegicus* observed in the ROV videos appear to be pregnant.

The morphology of the carbonates providing a protective shelter, and the site's elevated temperatures are said to possibly enhance reproductive success by accelerating egg development.

Lost fishing gear on the carbonate rock around the perimeter of the craters was observed by ROV imagery. The gear was heavily colonised by sessile fauna (e.g., anemones and hydrozoans), suggesting being old. No recent fishing activities (VMS and AIS, OSPAR/ICES) are recorded around the mud volcano system.

The presence of red-listed *S. norvegicus* and VME octocorals suggests that the Borealis is a sanctuary for endangered species.

The absence of bottom trawling and the natural habitat's protective qualities offer a refuge where these species can thrive despite significant seafloor impacts from fishing in the surrounding Barents Sea.

- Will the work be of significance to the field and related fields? How does it compare to the established literature? If the work is not original, please provide relevant references.

Despite numerous observations of methane emissions from the seafloor in Arctic regions, only five mud volcanoes have been discovered in the Canadian Beaufort Sea (Western Arctic), and three in Alaska, and so far, the Håkon Mosby Mud Volcano was the only known structure in Norwegian waters.

This work is significant to the field because it is an extremely important new and seldom discovery of a mud volcano that will need to be registered on sea-maps as soon as possible to avoid damaging this complex habitat and its associated fauna. Comparison to established literature is good, but there are issues that need to be taken care of.

- Does the work support the conclusions and claims, or is additional evidence needed?

Lines 215-218

Saithe (*Pollachius virens*), spotted wolffish (*Anarhichas minor*), cod (*Gadus morhua*), bearded rockling (*Enchelyopus cimbrius*) can be identified on images. But splitting redfish into *Sebastes norvegicus*, *S. viviparus* and *S. mentella* is challenging and would need more verification to become trustworthy (e.g., to be taken by physically by equipment and identified by experts). This would need to be explained in the text.

While we acknowledge the reviewer's suggestion that collecting specimens would have been ideal for species identification and confirming pregnancy, capturing redfish using the ROV was not feasible due to the size and mobility of the fish. Attempting to do so would

have been cruel and could have caused unnecessary suffering to the animals and potential damage to the ecosystem. Instead, we relied on the expertise of fish specialists from the University Museum of Bergen to assist with identifying the fish from the ROV footage, as noted in the acknowledgements section. We have revised the text to emphasize the uncertainty surrounding the identification of *Sebastes* spp. and have detailed the specific characteristics used for differentiation where possible.

Lines 223-225: Several individuals of the red-listed *S. norvegicus* observed in the ROV videos appear to be pregnant.

This is written without any references or further description. As above, this would need much more evidence since it is extremely difficult (impossible) to judge if a *S. norvegicus* is pregnant from a picture.

We thank the reviewer for the comment that allowed us to explain this part better, which is very important for our paper. We have expanded upon the assumption that *S. norvegicus* observed in the ROV videos appears to be pregnant, explaining that it is based on the observation of the fish's distended abdomens coupled with the timing of these observations (early May), which aligns with the known spawning season for this species as supported by publications indicated now in the main text. On this matter, our opinion diverged from the reviewer's opinion, and as explained above, instead, we relied on the expertise of fish specialists from the University Museum of Bergen to assist with identifying the fish from the ROV footage.

Lines (226-228) 287-290: The elevated temperatures around the seepage sites may create optimal conditions for reproductive processes, leading to more effective breeding cycles. This thermal advantage can enhance the survival and growth of fish populations in the deep sea, and the Barents Sea, one of the world's most productive fishing grounds, represents a significant ecological asset.

It is highly unlikely that a 500m diameter spot can have an effect on the “fish-population” in the Barents Sea. It may have an impact on the local fish using this area for spawning. This is written without any references or further evidence and need to be more carefully evaluated.

There have now been several observations of animals in the deep sea aggregating and spawning around seeps and vents (e.g. Drazen et al., 2003; Hartwell et al., 2018; Barry et al., 2023). The direct causation of such aggregations is not known for sure, however, several explanations have been offered. First, deep-sea organisms often exist at low population densities and so aggregation around prominent topographic features can be advantageous

for reproduction. The conditions around seeps provide both enhanced local productivity and higher than surrounding ambient temperatures. These offer enhanced feeding opportunities for both adult and juvenile stages of organisms. Elevated temperatures are likely to enhance the development times of eggs and larvae of organisms in areas of seepage, with Drazen et al. (2003) estimating an increase in temperatures of 1.5°C at the Gorda Ridge in the Pacific could reduce egg incubation time by 10% in benthic fish and octopus assuming a Q10 of 2. However, we recognize that the taxa observed at Borealis (e.g. the redfish) have a different reproductive ecology than the species in the referenced papers since they have a direct development releasing free-swimming larvae that might not be as affected by the increased temperature as eggs incubated directly on the seafloor. Still, the environment around the Borealis MV may provide both elevated food, temperature, an obvious “reef-like” effect, aggregating fish taking advantage of shelter and protection from bottom trawling. Given the conservation status of the fish in the Barents Sea, with some species threatened as a result of overexploitation, even a relatively small location like the *Borealis* MV and associated carbonates could be significant as a population refuge.

Lines 216-221: Large schools of fish are clustering around the jagged carbonate formations suggest that these structures serve as critical habitats, offering both shelter and abundant feeding opportunities, thereby playing a pivotal role in sustaining the local fish populations.

This is built on evidence from observations in Lophelia reefs and artificial reef-life structures – but I am missing evidence that the pelagic prey that the mention fish species are eating are found at the volcano structures. Did you register these pelagic prey-species on your video-records.

We thank the reviewer for the comment. This comment is connected to the previous comment, where we have explained the role of jagged carbonate formations, which provide habitat and feeding opportunities for large schools of fish and draw parallels with observations made in Lophelia reefs and artificial reef-like structures. However, we acknowledge that our manuscript did not explicitly document direct evidence of pelagic prey species associated with these carbonate formations at the Borealis MV. Additionally, it's important to consider that the feeding dynamics at these sites might involve a broader range of prey than previously observed in the gut content of fishes captured in the pelagic zone. *Sebastes* spp. have quite a variable diet composed largely of small fish and crustaceans, with relative contribution of specific food-items changing with size of the fish (Dolgov and Drevetnyak, 2011). The biologically enriched environment of Borealis MV

driven by the seep activity and associated bacterial production could be supporting a more complex food web that benefits the resident fish populations.

Lines 169-204

It is very hard to understand what was observed via ROV imagery, what was sampled using a push core, and how sediment was taken for DnA analyses. I question is it may be too premature to include the faunistic information in this paper since it anticipated that another and likely more describing paper will be made by the authors.

We thanked the reviewer and agreed that the text needed to be clarified regarding the analysed samples and how they were collected. We have added additional information in several paragraphs throughout the text to clarify which observations are from ROV imagery versus samples and information in Supplementary Table 1.

We value the reviewer's comment regarding the potential prematurity of including faunistic data in this paper. We are convinced that incorporating this faunistic information is crucial at this stage, as it establishes a baseline that highlights the unique ecological characteristics of this mud volcano in the Barents Sea. Given that this area is likely to be exploited in the future (in fact, it has been included in one of the drilling permits from the Norwegian Offshore Directorate), presenting this data now underscores the need for careful management to mitigate potential risks associated with its development. To avoid including preliminary identifications of fauna that are likely to contain errors that have to be corrected in the following papers, we have maintained a focus in this paper on the megafauna that could be identified with relative certainty based on ROV imagery and kept identifications of macrofauna to very broad categories. Still, we believe that the information provided in the paper is sufficient at this stage to support our conclusions and hypotheses about the importance of the Borealis faunal communities for local productivity and biodiversity.

Lines 187-192: Anemones, annelids, amphipods, gastropods, polyplacophorans, and ophiuroids.

These large animal groups are found everywhere and does not really provide any information. I may suggest describing their traits or morphological characteristic in this paper instant, and then in next paper provide more detailed taxonomy.

While we agree with the reviewer that the faunal groups identified is not very informative, more detailed information on the taxonomy or traits of the collected fauna is not possible to supply at this point as the faunal samples are still under processing. As described

above, we kept identifications at higher taxonomic levels at this stage to avoid including preliminary identifications with a likelihood of error. Identifications to lower taxonomic levels (genus and species) were only included where we were quite confident they were correct.

- Are there any flaws in the data analysis, interpretation and conclusions? - Do these prohibit publication or require revision?

Some sentences need to be strengthened and definitions clarified before publication:
Lines 352-374

A photogrammetry technique (SfM) was applied and selected ROV video frames provided physical habitats and sedimentary facies. But are these pictures also analyzed for fauna – please explain?

We thank the reviewers for the comments and recognizing that our initial explanation might not have been sufficiently clear. In our study, we utilized the same video footage for multiple purposes: to construct the mosaic presented in Figure 2 using the Structure from Motion (SfM) technique and to perform detailed video analyses for seafloor characterization and fauna annotation that gave us the possibility to estimate the abundance of redfish and corals as shown in Figure 4. These methodologies are now better explained in Chapter 3 "ROV and Video Analyses".

The substrate was differentiated in carbonate crusts (CC), Microbial Mats (MM) and all other substrate (denoted as BG in figure 4, but not as such in the text, and this is confusing).

Macrofauna occurrences were classified as Red Fishes (RedF) and Coral (CC). The Coral category are denoted "CC" in line 366 (but not in Figure 4) -the same as carbonate crusts – and this is confusing.

We thank the reviewer for this comment, which allowed us to refine and clarify this part in our manuscript. We have now detailed the various substrates identified in the study area through video analysis in the section titled "A Unique Oasis for Faunal Communities." Additionally, we have ensured consistency in the terminology used for different substrate analyses across the main text, the tables, and Figure 4.

Each ROV dive surveyed different areas, with some level of overlapping between ROV tracklines crossing the Borealis MV main seeping sites. – **Does this represent a danger of counting and recording of observations more than ones?**

In response to the reviewer's concern, we would like to emphasise that the primary goal of our study was to spatially visualise associations among observations rather than performing statistical analyses on fauna abundance. However, the reviewer raises a valid point regarding the potential issues in recording observations, especially for mobile organisms like fish. From our observations during the expedition and the following video analyses, we noted that the redfish has minimal movement, often remaining stationary around the carbonate crusts. This behaviour reduces the probability of double-counting these particular fish in our recordings. We also observed other types of fish, and those were more mobile than the redfish. This is why we decided not to include them in our classification. As for the corals, each coral observed was precisely pinpointed and georeferenced using the ROV's navigation system on the tracklines. These locations were marked as waypoints on our maps during the dives, and corrected once we performed the observations of the videos onshore. This method allowed us to maintain a reliable record of coral positions, avoiding the danger of recording the exact coral multiple times.

The surveyed area covered was estimated by using GIS-based tool and applying a 1 m² buffer area surrounding each ROV position. Because of the differences in the extent of the areas surveyed by each video, the abundance of the detected macrofauna (Red Fishes and Coral) are expressed as number per 10 square meter.

- Is the methodology sound? Does the work meet the expected standards in your field?

The analyzing of fauna from video (and maybe images?) is an accepted and well used method. The abundance of observed fauna is standardized. But the paper would be clearer and more convincing if explained and verified more deeply (see above). I guess that this will be the case for the next paper wish I will look very much forward to.

We have provided a more precise and better explanation of the methods used to analyse our data and samples as better explained above.

- Is there enough detail provided in the methods for the work to be reproduced?

There are not enough details provided for the fauna-work to be reproduced. I guess that this will be the case for the next paper wish I will look very much forward to.

As described above, additional details on the processing of faunal samples were added to the methodology section, and further information about the fauna samples collected has been added to Table 1 in the supplementary material.

On behalf of the co-authors

Giuliana Panieri

November 18, 2024